# DATA INTERPRETER:
# AN LLM AGENT FOR DATA SCIENCE

## ABSTRACT

Large Language Model (LLM)-based agents have shown effectiveness across many applications. However, their use in data science scenarios requiring solving long-term interconnected tasks, dynamic data adjustments and domain expertise remains challenging. Previous approaches primarily focus on individual tasks, making it difficult to assess the complete data science workflow. Moreover, they struggle to handle real-time changes in intermediate data and fail to adapt dynamically to evolving task dependencies inherent to data science problems. In this paper, we present **Data Interpreter**, an LLM-based agent designed to automatically solve various data science problems end-to-end. Our Data Interpreter incorporates two key modules: 1) *Hierarchical Graph Modeling*, which breaks down complex problems into manageable subproblems, enabling dynamic node generation and graph optimization; and 2) *Programmable Node Generation*, a technique that refines and verifies each subproblem to iteratively improve code generation results and robustness. Extensive experiments consistently demonstrate the superiority of Data Interpreter. On InfiAgent-DABench, it achieves a 25% performance boost, raising accuracy from 75.9% to 94.9%. For machine learning and open-ended tasks, it improves performance from 88% to 95%, and from 60% to 97%, respectively. Moreover, on the MATH dataset, Data Interpreter achieves remarkable performance with a 26% improvement compared to state-of-the-art baselines. Code will be open-sourced upon publication.

## 1 INTRODUCTION

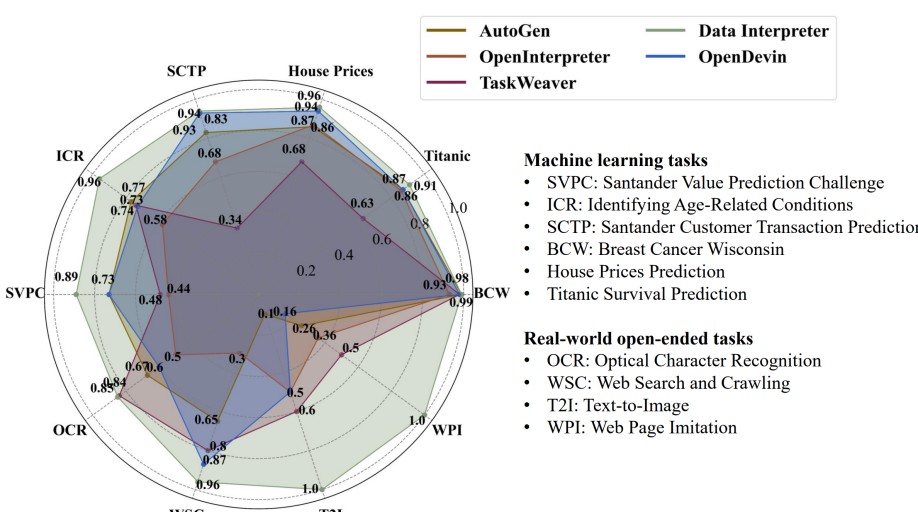

Figure 1: Comparison across various open-source frameworks on various data science tasks. We define a unified metric, the *comprehensive score* (Appendix D.2.), to standardize performance evaluation across tasks with different metrics. A higher score indicates better performance.

Large Language Models (LLMs) have demonstrated remarkable adaptability across a wide range of applications, excelling in areas such as software engineering (Hong et al., 2023), open-world

navigation (Wang et al., 2023a;b;c; Chen et al., 2024), collaborative intelligence (Zhuge et al., 2023; 2024; Zhang et al., 2024a), and scientific research (Tang et al., 2024). However, their performance in data science remains limited.

Data science (De Bie et al., 2022; Hassan et al., 2023), the practice of extracting insights from data, spanning from data gathering to model building and decision-making. It integrates multiple disciplines such as computer science, statistics, data visualization, and mathematics (Zhang et al., 2023). As discussed in (Zhang et al., 2024c; Zheng et al., 2021), data science workflows are inherently complex, involving interconnected tasks such as data processing, feature engineering, and model training. Solving these tasks requires iterative refinements and real-time adjustments, as both data and requirements continuously evolve.

Leveraging the extensive knowledge and coding capabilities of LLMs, recent efforts (Shen et al., 2024; Hollmann et al., 2023; Bordt et al., 2024; Zhang et al., 2024c; Liu et al., 2024) have integrated LLMs into data science tasks. These approaches primarily focus on individual tasks, such as feature engineering (Hollmann et al., 2023), model selection (Shen et al., 2024), and hyperparameter optimization (Liu et al., 2024), often operating within fixed pipelines. However, they lack a holistic evaluation of end-to-end workflows, making it difficult to assess the complete data science process. Furthermore, these methods often struggle to handle real-time changes in intermediate data and adapt dynamically to evolving task dependencies. While recent works (Wu et al., 2023b; Zhang et al., 2023) have improved performance in data-related tasks, they remain inadequate for machine learning or comprehensive data transformation tasks, involving intricate task interdependencies that require continuous updates and dynamic global planning (Zhang et al., 2024c).

To address these challenges, we present **Data Interpreter**, an LLM agent that reframes the data science workflows as a *Hierarchical Graph Modeling* problem, where interconnected tasks are represented as nodes, and their dependencies as edges within the graph. This structured representation enables dynamic and flexible task management, allowing the system to adjust to evolving data and task requirements in real-time, and thus efficiently manages the complex, interdependent steps of data science. Another core of Data Interpreter is *Programmable Node Generation*, a key innovation that automates the real-time generation, refinement, and verification of nodes in the graph. This ensures that each subproblem is accurately defined and executed, improving the robustness and precision of the workflow. Leveraging the coding capabilities of LLMs, the system dynamically synthesizes and optimizes the graph structure, making it highly adaptable to the demands of complex, evolving data science tasks.

Our experiments demonstrate that Data Interpreter significantly outperforms existing methods across several benchmarks, achieving a 25% performance boost on the public dataset InfiAgent-DABench, and a 26% improvement on the MATH dataset. Compared to other open-source frameworks, Data Interpreter consistently shows notable advancements in machine learning and open-ended tasks, as illustrated in Figure 1. By rethinking how data science workflows are structured and managed, Data Interpreter sets a new standard for adaptability and efficiency, offering a powerful solution for complex, real-world applications.

## 2 RELATED WORK

**LLMs as Data Science Agents** Large language models (LLMs) have demonstrated expert-level knowledge in machine learning and have made significant progress in automating data science tasks. Early research focused on using LLMs to write code, aiming to simplify complex computations involved in reasoning processes (Gao et al., 2023; Chen et al., 2022). Subsequent work introduced code interpreters that leverage function-calling mechanisms, offering greater flexibility in solving complex problems (Zhou et al., 2023; Gou et al., 2024; Wang et al., 2024a). This interpreter-based approach has now become a mainstream method for enabling LLMs to handle complex reasoning and scientific tasks (Huang et al., 2023b; Hassan et al., 2023; Qiao et al., 2023; Zhang et al., 2024b). Recently, Zhang et al. (2023) introduces an LLM-based agent for data analysis, demonstrating capabilities in data processing and exploration within a code-centric framework, but does not evaluate its performance on predictive tasks such as machine learning pipelines. Guo et al. (2024) harness LLMs and case-based reasoning to solve data science tasks, leveraging human expertise to enhance the efficiency of LLM-based agents in data science, which is complementary to our work. Liu et al. (2024) uses LLMs to perform hyperparameter tuning to automate machine learning tasks focusing on

single task rather than full pipeline construction and evaluation. Therefore, end-to-end evaluation frameworks specifically designed for data science tasks remain insufficiently developed. To address this gap, we propose a unified, general framework specifically designed for data science tasks. Our framework has been rigorously benchmarked across diverse tasks and settings, offering valuable insights into the application and effectiveness of LLMs in data science.

**Enhancing LLM with Tools** Recent research has focused on enhancing LLM capabilities by integrating external tools (Schick et al., 2024; Paranjape et al., 2023). Zhuge et al. (2023); Shen et al. (2024) introduced multi-agent systems to tackle multimodal tasks, while Yuan et al. (2023); Liu et al. (2023) proposed frameworks for retrieval and automatic tool selection, eliminating the need to assign tools for specific tasks statically. Recent efforts have increasingly focused on integrating tool-using abilities into a structured pipeline, enabling sophisticated task planning, tool invocation (Wu et al., 2023a; Shen et al., 2024; Liang et al., 2024). Qian et al. (2023); Yuan et al. (2024) discuss the creation and instruction of the tool from code-form or lengthy tool documentation to enhance tool utilization efficiency. In this paper, we further advance these ideas by enabling LLMs to dynamic orchestration and combination of multiple tools. Our approach improves practicality by leveraging execution experience, allowing LLMs to select and combine tools as needed independently.

**Graph-Based Planning for LLM Agents** Planning is a critical capability of LLM-based agents, focusing on generating logically structured action or thought roadmaps for specific problems (Huang et al., 2024b; Chen et al., 2024). Earlier works like CoT (Wei et al., 2022; Yao et al., 2022) decompose complex tasks into subtasks and perform sequential planning. However, due to the complexity of certain problems, a single plan generated by an LLM-based agent is often insufficient. To address this, ToT (Yao et al., 2024) and GoT (Besta et al., 2023) introduce automatic tree or graph structures that refine node-level LLM prompts, optimizing connectivity to improve performance. Similarly, DSPy (Khattab et al., 2023) abstracts LLM pipelines as text transformation graphs, while PRODIGY (Huang et al., 2023a) applies graph-based in-context learning and pre-training methods. Further, Zhuge et al. (2024) enhance node prompts and agent coordination via graph connectivity adjustments, and Vierling et al. (2024) develop a learnable model to dynamically generate edges between agents in a graph, facilitating internal communication. While these planning approaches excel in various domains, they often struggle with multi-step, task-dependent problems commonly encountered in data science. In this paper, we explore the potential of integrating graph structures with LLM-based agents for data science tasks—an area that remains largely untapped despite emerging related work.

## 3 METHODOLOGY

In this section, we first present the foundational formulation of hierarchical graph modeling for data science problems, defining the task graph and action graph in Section 3.1. Next, we detail the iterative optimization process of the hierarchical graph structure in Section 3.2. Finally, in Section 3.3, we introduce programmable node generation, explaining how we integrate expertise at different granularities to improve the performance of LLMs.

### 3.1 HIERARCHICAL GRAPH MODELING FOR COMPLEX TASK DECOMPOSITION

Data science problems, particularly those involving machine learning, encompass extensive detailing and long-range workflows, including data pre-processing, feature engineering, and model training. This long-term planning complicates the direct planning of all detailed tasks and coding. Drawing inspiration from the application of hierarchical planning in automated machine learning tasks (Mohr et al., 2018; Mubarak & Koeshidayatullah, 2023), we organize the data science workflow via hierarchical structure, which initially decomposes the intricate data science problem into manageable tasks and further break down each task into specific actions executed through code (see Figure 2).

Therefore, solving a data science problem can be formulated as follows: given a task-oriented input $x$, we seek to apply a series of operators, unified as a function $\mathtt{P}$, to produce an output $\hat{y} = \mathtt{P}(x)$. Our goal is for $\mathtt{P}$ to generate solutions that closely approximate or match the anticipated $y$. However, due to the complexity of $\mathtt{P}$, which may involve various operations and intermediate data, fully automating the solution to a task is typically challenging.

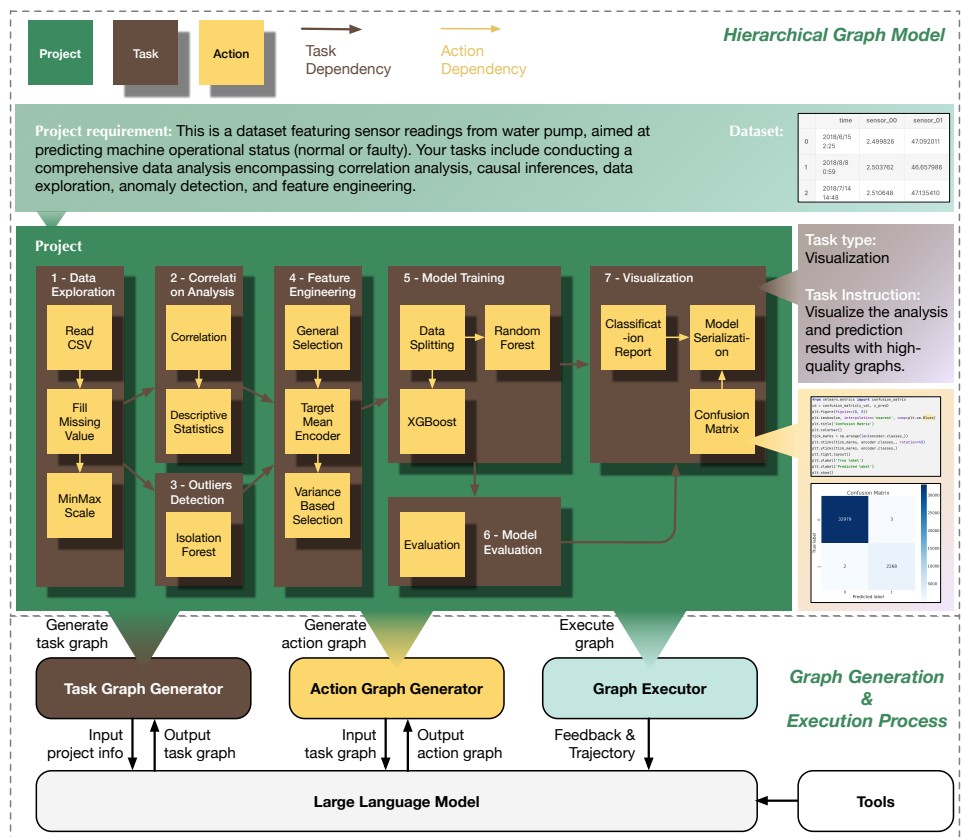

Figure 2: **Data Interpreter Example Workflow.** The upper section illustrates how Data Interpreter organizes a data science workflow using a hierarchical structure. The process begins by decomposing project requirements into a task graph, which is then further broken down into actions executed through code. The lower section highlights the core modules of Data Interpreter, including the *task graph generator*, *action graph generator*, and *graph executor*. These modules work together to manage task execution and provide real-time feedback. The *graph executor* efficiently executes the action graph using reflection and integrated tools, delivering essential real-time feedback throughout the process.

Leveraging the reasoning ability of LLMs for general task decomposition, our method decomposes the solving process of $\mathrm{P}$ into a series of sub-processes $\mathrm{p}_1, \mathrm{p}_2, \mathrm{p}_3, \dots$ that can be directly solved and verified. The primary challenge lies in determining the relationships $r = \langle \mathrm{p}_i, \mathrm{p}_j \rangle \in \mathcal{R}$ between these sub-processes. Our framework represents all subprocesses as nodes within $\mathrm{P}$, ultimately forming a graph $\mathcal{G}$ that embodies the entire function $\mathrm{P}$:

$$\hat{y} = \mathcal{G}\left(\{\mathrm{p}_i(x)\}_{i=1}^{n}, \mathcal{R}\right), \tag{1}$$

where $\mathcal{G}$ represents a Directed Acyclic Graph (DAG) composed of the sub-functions $\mathrm{p}_1, \mathrm{p}_2, \mathrm{p}_3, \dots$ interconnected through the relationships $\mathcal{R}$. This graph illustrates how these sub-functions are combined to generate the final output $\hat{y}$. Unlike traditional reinforcement learning (RL) methods for planning (Moerland et al., 2023; Schmidhuber, 2003), which often require a substantial number of demonstrations to perform domain-specific training, our approach leverages the in-context learning of LLMs. This training-free nature allows our method more adaptable and efficient for general task decomposition.

Improving $\mathcal{R}$ involves achieving an optimal node topology, which has demonstrated robust performance and flexibility in prior research Zhuge et al. (2024). In our framework, all subprocesses exchange intermediate results and parameters, represented as $r = \langle \mathrm{p}_i, \mathrm{p}_j \rangle \in \mathcal{R}$. Given the inherent challenges in data science problems Hutter et al. (2019), this process can be complex. However, we

can optimize the graph topology by refining the relationships between subprocesses. Our objective is:

$$\mathcal{G}^* = \arg\max_{\mathcal{G}} \; \mathbb{E}_{x \sim \mathcal{D}} \left[ \text{Performance} \left( \mathcal{G} \left( \{ \mathrm{p}_i(x) \}_{i=1}^n, \mathcal{R} \right), y \right) \right], \quad (2)$$

where $\mathbb{E}_{x \sim \mathcal{D}}$ denotes the expectation over the data distribution $\mathcal{D}$, and Performance measures the accuracy of the predicted output $\hat{y}$ against the target output $y$. Importantly, within $\mathcal{G}^*$, if a subprocess $\mathrm{p}_i$ proves challenging to solve, it can be further decomposed into smaller, more manageable subprocesses. Next, we will illustrate the core concepts in our hierarchical graph modeling with an example.

**Task Graph.** Data Interpreter utilizes LLMs to perform task planning, providing only the project requirement as the goal without relying on pre-defined steps, tasks and relationships. As shown in Figure 2, an example workflow decomposed by Data Interpreter for a machine operational status prediction problem, might include tasks like: `data exploration`, `correlation analysis`, `outliers detection`, `feature engineering`, `model training`, `model evaluation`, and `visualization`. Each task node is defined within the metadata and includes attributes such as task description, task type, status, execution feedback, and dependencies, collectively form the task-level graph $\mathcal{G}$, enabling structured task management and execution. Consequently, during the solving process, the dynamic contextual data are automatically constructed and acquired through the inter-dependencies among tasks, avoiding the need to retrieve the entire context at once while maintaining the relevance of the input context, offering flexibility and scalability for broader data science applications.

**Action Graph.** Data Interpreter breaks down each task into multiple actions using contextual memory, thus forming an action graph. Action graphs can be executed and verified independently, and the synthesis of each action node will be detailed in Section 3.3. As illustrated in Figure 2, the visualization task is divided into three distinct actions, with the confusion matrix calculation handled by `sklearn`. The solving process is represented as an action graph, visually captures the relationships between these actions and serves as an implicit representation of the code. Additional runtime examples are provided in Figure 7 in the Appendix.

At finer granularity, action graph iteratively adjusts to handle real-time execution feedback, such as managing failures by refining code or incorporating verification processes, making it a sufficiently granular unit for rapid task adjustments and validation. We explore this optimization process further in Section 3.2.

### 3.2 TASK GRAPH: ITERATIVE GRAPH REFINEMENT

**Task Graph Generation and Execution.** A key advantage of our approach is its ability to dynamically adjust the task graph in response to changing environments, unlike prior methods (Wei et al., 2022; Besta et al., 2023; Yao et al., 2022) and frameworks such as OpenInterpreter (Lucas, 2023) and AutoGen (Wu et al., 2023b), which generate static plans for one-time execution. Our method introduces iterative graph optimization, allowing it to adapt to a dynamic environment through continuous updates.

As shown in Figure 2, Data Interpreter uses a task graph generator to initialize the task graph as discussed in Section 3.1. Each task is then translated into executable code by the action graph generator, which takes into account the outcomes of prior tasks to ensure contextual consistency. The generation process is detailed in Algorithm 1.

To ensure runtime verification and provide real-time feedback during execution, Data Interpreter incorporates a stateful graph executor that manages both execution and debugging using reflection mechanisms (Shinn et al., 2024). Specifically, if the execution encounters exceptions or fails a verification check, the action graph generator dynamically reflects on the execution results, and then regenerates the code to resolve the issue or optimize the output, providing data-driven feedback. This process is collectively conducted by action graph generator and graph executor.

**Task Graph Refinement.** The task graph generator manages tasks, monitors their statuses and dependencies, and dynamically adjusts the task graph by adding, removing, or modifying tasks as needed. Each task is further decomposed into an action graph, which consists of one or several action nodes. Each action graph can be executed and evaluated independently, allowing for granular control and flexibility in the execution process.

---

**Algorithm 1** Iterative Graph Execution

---

**Input:** User requirements $req$, large language model $LLM$, tool sets $T$
**Output:** Optimized graph $G^*$

1: Set $M$ as the maximum number of iterations, $R$ to denote runtime results
2: $G \leftarrow$ initialize_graph$(req, LLM)$         ▷ Initialize the graph with user requirements
3: **while** not $G$.is_finished() **do**       ▷ Iterative process until termination condition is met
4:     $tn \leftarrow$ select_task_node$(G, LLM)$    ▷ Monitor task execution and select the next task node
5:     $ag \leftarrow$ initialize_action_graph$(tn, T, LLM)$       ▷ Generate codes based on task node
6:     **for** $i = 1$ to $M$ **do**         ▷ Execute up to M iterations or until success
7:        $R \leftarrow$ execute$(ag)$     ▷ Execute the action graph and return runtime results
8:        **if** is_success$(R)$ **then**        ▷ Determine if execution success or not
9:           **break**          ▷ Exit loop if the action is successful
10:        **end if**
11:        $ag \leftarrow$ refine$(tn, R, LLM)$        ▷ Refine the action graph based on runtime result
12:     **end for**
13:     $tn \leftarrow$ update_node_state$(tn, ag, R)$       ▷ Update the state of the task node
14:     $G$.task_graph $\leftarrow$ update_task_graph$(G, tn)$     ▷ Integrate updates into the task graph
15: **end while**
16: $G^* \leftarrow$ finalize_graph$(G)$         ▷ Save optimized graph
17: **return** $G^*$

---

During execution, a task is marked as `Success` if the corresponding code executes successfully. If execution fails, Data Interpreter leverages LLMs to debug the code based on runtime errors, making up to a predefined number of attempts to resolve the issue. If the problem persists after the set attempts, the task node is flagged as `Failure`, as shown in Figure 3.

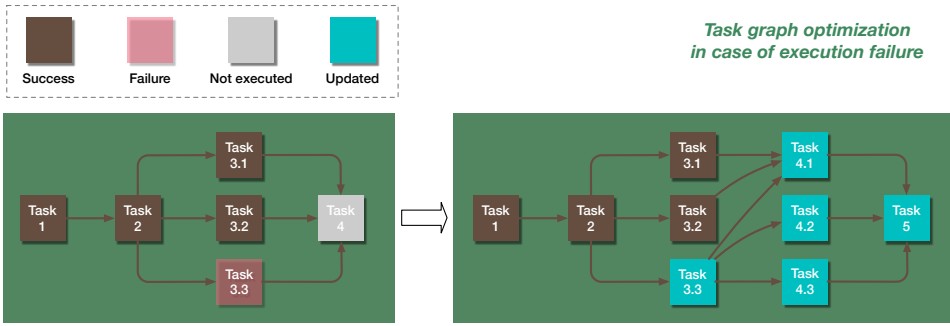

Figure 3: **Task Graph refinement of Data Interpreter.** Task graph refinement for the failed task. After task execution, Task 3.3 fails. The refined task graph integrates existing success tasks, replaces task 3.3 with the updated task 3.3, and introduces new tasks 4.1, 4.2, 4.3 and 5.

For failed tasks, Data Interpreter regenerates the task graph based on current episodic memory and the execution context, as depicted in Figure 3. Given the task dependencies, the regenerated task graph is sorted topologically and compared to the original using a prefix matching algorithm (Waldvogel, 2000) to identify differences in task descriptions. This comparison helps identify divergence points (forks), and the final output includes all unchanged tasks before the fork, along with any new or modified tasks after the fork. This approach allows Data Interpreter to efficiently locate the parent node of the failed task and seamlessly integrate the newly generated task and its subsequent tasks into the original graph. It directly leverages the completed memory of all dependent tasks during re-execution, avoiding unnecessary code regeneration or redundant executions.

By employing continuous monitoring and iterative updates, Data Interpreter avoids the inefficiencies associated with generating all tasks upfront. This dynamic adjustment of both the code and planning levels based on task outcomes enables modifications at varying levels of granularity, significantly improving overall efficiency.

### 3.3 Action Graph: Programmable Node Generation

**Action Node.** An action node, as introduced in Section 3.1, represents an executable code snippet that encapsulates the computational logic required for task execution. Each action node can encompass data transformations, function calls, or other relevant operations, making it the fundamental unit of execution within the action graph. It integrates both external functions and operators invoked from various tools, as well as non-tool logic derived from libraries such as Pandas and NumPy. By combining tool-based operations and library functions into a single executable code snippet, action nodes ensure uniform and flexible execution across different tasks.

**Tool Selection.** Effective tool selection and integration, particularly in the context of task-specific requirements, play a crucial role in the success of task execution, as noted in prior research (Qian et al., 2023; Yuan et al., 2024; Huang et al., 2024a; Liu et al., 2023). In Data Interpreter, we leverage task dependencies to enrich the task-specific context, thereby enhancing the decision-making process for tool selection and code generation.

During the execution of each task $p_i \in \mathcal{G}$, where $\mathcal{G}$ represents the task graph, Data Interpreter first retrieves suitable tools before generating the associated code. The task metadata $q(p_i)$, which includes textual information such as task descriptions and types as well as graph-structured task dependencies, is used as a query to retrieve a list of candidate tools from the available toolset $T = \{t_1, t_2, \ldots, t_n\}$. The model ranks these tools by evaluating their semantic relevance to the task using their functionality schemas $\mathcal{S}(t_j)$. This produces a ranked list $R(p_i, T) = \{r_1, r_2, \ldots, r_n\}$, where each tool $t_j$ is ranked according to its suitability for the task. From this ranked list, Data Interpreter selects the top-$k$ tools, denoted as $T_k(p_i) \subseteq T$, to assist in executing task $p_i$. Importantly, Data Interpreter can bypass tool selection when no suitable tools are found, relying solely on the LLM to generate appropriate code. This flexibility ensures that the system can adapt to a wide range of task requirements without being restricted by tool availability.

**Programmable Node Generation.** Unlike conventional LLM-based agent frameworks that invoke tools through isolated function calls, Data Interpreter generates comprehensive code snippets that seamlessly integrate selected tools within the broader logic of the task. Based on the tools selected from $T_k(p_i)$, Data Interpreter dynamically incorporates them into the code, aligning their functionality with the specific task context. This approach allows tools to function in the same manner as standard libraries like NumPy, enabling adaptive tool usage that adjusts to evolving task conditions. For example, in the deployment workflow, the CatCount tool dynamically utilizes its fit and transform functions depending on the task context, as illustrated in Figure 6 in the Appendix.

Our programmable node generation approach not only ensures that tools are used in a context-aware and task-specific manner but also facilitates the seamless integration of domain-specific expertise. By allowing real-time adaptability and optimization of tool usage, Data Interpreter significantly enhances the efficiency and robustness of task execution, representing a novel contribution to LLM-based task automation.

## 4 Experiments

### 4.1 Experimental setup

**InfiAgent-DABench:** InfiAgent-DABench (Hu et al., 2024) evaluates LLMs in data analysis tasks across 257 challenges from 52 CSV files, covering 7 categories: summary statistics, feature engineering, correlation analysis, machine learning, distribution analysis, outlier detection, and comprehensive data preprocessing. We used accuracy as the evaluation metric. Data Interpreter was primarily evaluated with `gpt-4o` and `gpt-4-0613` (temperature=0), and compared against XAgent (Team, 2023), AutoGen (Wu et al., 2023b), as well as other baselines reported from (Hu et al., 2024).

**ML-Benchmark:** To evaluate the performance of solving real-world machine learning challenges, We collected 8 datasets from Kaggle for ML-Benchmark (details in Table 13. We also detailed the evaluation metrics on ML-Benchmark in Appendix D.2. Baselines included XAgent, AutoGen, OpenInterpreter (Lucas, 2023), TaskWeaver (Qiao et al., 2023), and OpenDevin (Wang et al., 2024b). As default, we used `gpt-4-1106-preview` with temperature set to 0.

Table 1: **Performance comparisons on InfiAgent-DABench.** Results marked with an asterisk (*) are reported by Hu et al. (2024). Rows marked with a dagger symbol (†) indicate the w/o Agent baseline for comparison. The $\Delta$ column represents the accuracy improvement of the agent framework compared to the w/o agent setups. The best results are highlighted in bold.

| Agent Framework | Model | Accuracy (%) | $\Delta$ (%) |
|---|---|---|---|
| w/o Agent | gemini-pro | 56.42* | - |
| | gpt-3.5-turbo-0613 | 60.70* | - |
| | gpt-4-0613 | 78.99*† | - |
| | gpt-4-0613 | 75.21 | - |
| | gpt-4o | 75.92† | - |
| XAgent | gpt-4-0613 | 47.53* | -31.46 |
| AutoGen | gpt-4-0613 | 71.49 | -7.50 |
| Data Interpreter | gpt-4-0613 | 73.55 | -5.44 |
| Data Interpreter | gpt-4o | **94.93** | **+19.01** |

Table 2: **Performance comparisons on ML-Benchmark.** This table reports the comprehensive score of each task. "WR", "BCW", "ICR", "SCTP", and "SVPC" represent "Wine recognition", "Breast cancer wisconsin", "ICR - Identifying age-related conditions", "Santander customer transaction prediction", and "Santander value prediction challenge", respectively.

| Model / Task | WR | BCW | Titanic | House Prices | SCTP | ICR | SVPC | Avg. | Cost ($) |
|---|---|---|---|---|---|---|---|---|---|
| AutoGen | 0.96 | **0.99** | 0.87 | 0.86 | 0.83 | 0.77 | 0.73 | 0.86 | - |
| OpenInterpreter | **1.00** | 0.93 | 0.86 | 0.87 | 0.68 | 0.58 | 0.44 | 0.77 | - |
| TaskWeaver | **1.00** | 0.98 | 0.63 | 0.68 | 0.34 | 0.74 | 0.48 | 0.69 | **0.37** |
| XAgent | **1.00** | 0.97 | 0.42 | 0.42 | 0 | 0.34 | 0.01 | 0.45 | 20.09 |
| OpenDevin | 0.98 | 0.98 | 0.87 | 0.94 | 0.93 | 0.73 | 0.73 | 0.88 | 3.01 |
| **Data Interpreter** | 0.98 | **0.99** | **0.91** | **0.96** | **0.94** | **0.96** | **0.89** | **0.95** | 0.84 |

**Open-ended task benchmark:** To verify the capability for dynamic data handling, we also crafted the Open-ended task benchmark comprising 20 tasks. Details about datasets are in the Appendix D.1. We adopted AutoGen and OpenInterpreter and OpenDevin as baselines with average results reported over three runs. We adopted `gpt-4-1106-preview` with temperature set to 0.

**MATH:** We evaluated 4 categories (C.Prob, N.Theory, Prealg, Precalc) of level-5 problems from the MATH dataset (Hendrycks et al., 2021), following the setting of (Wu et al., 2023c). Level-5 problems were chosen for their complexity and the challenges in reliable numeric interpretation. We used MathChat (Wu et al., 2023c) and AutoGen (Wu et al., 2023b) as baselines for the MATH benchmark.

## 4.2 MAIN RESULT

**Performance on InfiAgent-DABench.** As demonstrated in Table 1, with `gpt-4-0613`, Data Interpreter achieved a score of 73.55, outperforming AutoGen by 2.9%. Notably, it still did not surpass the performance of directly invoking the LLM. We found this is primarily due to the growing context overhead in the problem-solving process, where the context length exceeds the maximum window size of `gpt-4-0613`, leading to task failures. However, by incorporating LLMs like `gpt-4o` with longer context windows, Data Interpreter demonstrated outstanding performance, improving results by 25% compared to direct LLM inference. This indicates that Data Interpreter significantly enhances the LLM's multi-step reasoning capabilities across a wide range of data analysis tasks, especially as the number of interaction rounds increases and the context overhead grows.

**Performance on ML-Benchmark.** As shown in Table 2, Data Interpreter achieved a comprehensive score of 0.95 across tasks, outperforming AutoGen (0.86) and OpenDevin (0.88) by 10.3% and 7.9%, respectively. It was the only framework to achieve a score above 0.9 on tasks such as Titanic, House Prices, SCTP, and ICR. Additionally, the Data Interpreter demonstrated a significant advantage over other frameworks, with improvements of 31.5% and 21.9% over OpenDevin on the ICR and SVPC

Table 3: **Performance comparisons on Open-ended task benchmark.** This table reports the completion rate of each task. The tested tasks include "OCR" (Optical Character Recognition), "WSC" (Web Search and Crawling), and "ER" ( Email Reply), "WPI" (Web Page Imitation), "IBR" (Image Background Removal), "T2I" (Text-to-Image), "I2C" (Image-to-Code) and "MGG" (Mini Game Generation).

| Model / Task | OCR | WSC | ER | WPI | IBR | T2I | I2C | MGG | Avg. | Cost ($) |
|---|---|---|---|---|---|---|---|---|---|---|
| AutoGen | 0.67 | 0.65 | 0.10 | 0.26 | **1.00** | 0.10 | 0.20 | 0.67 | 0.46 | - |
| OpenInterpreter | 0.50 | 0.30 | 0.10 | 0.36 | **1.00** | 0.50 | 0.25 | 0.20 | 0.40 | - |
| OpenDevin | 0.60 | 0.87 | 0.10 | 0.16 | **1.00** | 0.50 | 0.80 | 0.90 | 0.60 | 1.41 |
| **Data Interpreter** | **0.85** | **0.96** | **0.98** | **1.00** | **1.00** | **1.00** | **1.00** | **0.93** | **0.97** | **0.41** |

tasks, respectively. Notably, Data Interpreter solved the tasks more efficiently, achieving an average score of $ 0.84 while operating at only 27.9% of OpenDevin's cost. Data Interpreter consistently completed all mandatory processes across datasets, maintaining superior performance. Further details can be found in Table 6 in the Appendix.

**Performance on Open-ended tasks.** Table 3 illustrates that the Data Interpreter achieved a completion rate of 0.97, marking a substantial 110.8% improvement compared to AutoGen and 61.7% improvement compared to OpenDevin. In OCR-related tasks, the Data Interpreter maintained an average completion rate of 0.85, outperforming AutoGen, OpenInterpreter OpenDevin by 26.8%, 70.0% and 41.7%, respectively. In tasks requiring multiple steps and utilizing multimodal tools/interfaces, such as WPI, I2C, and T2I, the Data Interpreter emerged as the sole method to execute all steps. Baseline frameworks failed to log in and obtain the status for the ER task, resulting in a lower completion rate. In contrast, Data Interpreter dynamically adjusted to task requirements, achieving a completion rate of 0.97.

**Performance on math problem.** As illustrated in the Figure 4, Data Interpreter achieved the best results across all tested categories, reaching 0.82 accuracy in the N.Theory category, marking a 0.16 improvement over the performance of AutoGen. In the most challenging category, Precalc, Data Interpreter obtained an accuracy of 0.29, an increase of 0.17 compared to AutoGen. On average, our Data Interpreter showed 26.5% relative improvement compared to AutoGen.

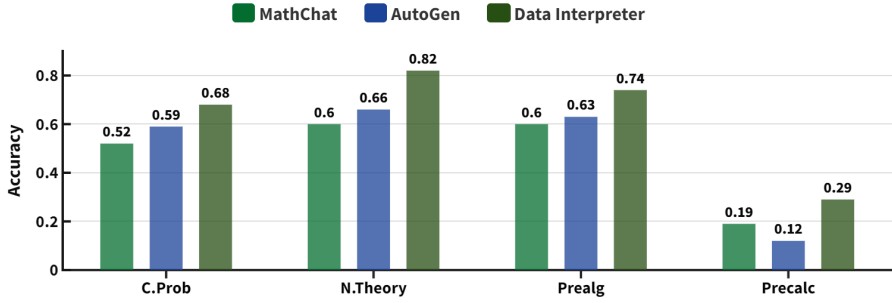

Figure 4: **Performance on the MATH dataset.** We evaluate all the problems with difficulty level 5 from 4 categories of the MATH dataset.

### 4.3 ABLATION STUDY

**Ablation on core modules.** We conducted ablation experiments with three configurations on the ML-Benchmark. First, we used ReAct (Yao et al., 2022) for code execution with simplified prompts, followed by the addition of iterative graph refinement, and finally, programmable node generation was introduced, using the Data Interpreter as the default. As shown in Table 4, iterative graph refinement improved performance by 0.48, enhancing dataset preparation and real-time tracking. Programmable node generation further boosted the comprehensive score by 10.6%, reaching 0.94. We detailed the results in Table 12.

Table 4: **Ablation on core modules.** Evaluated with Comprehensive Score on ML-Benchmark. "IGR" stands for Iterative Graph Refinement, and "PNG" denotes Programmable Node Generation. "ICR", "SCTP", and "SVPC" represent "ICR - Identifying age-related conditions", "Santander customer transaction prediction", and "Santander value prediction challenge", respectively.

| Code execution | IGR | PNG | House Prices | SCTP | SVPC | ICR | Avg. |
|:---:|:---:|:---:|:---:|:---:|:---:|:---:|:---:|
| ✓ | | | 0.51 | 0.17 | 0.66 | 0.17 | 0.37 |
| ✓ | ✓ | | 0.96 | 0.91 | 0.80 | 0.74 | 0.85 |
| ✓ | ✓ | ✓ | **0.96** | **0.95** | **0.89** | **0.96** | **0.94** |

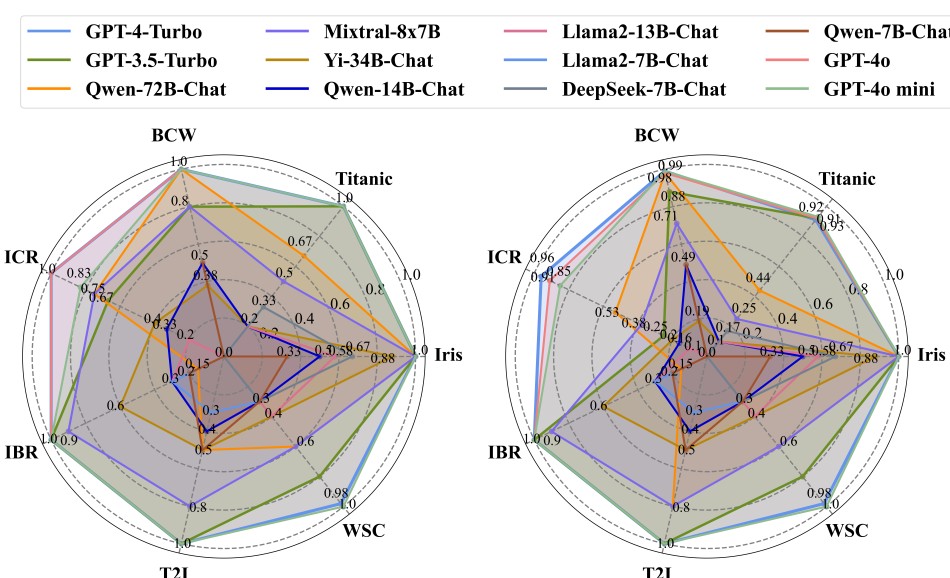

Figure 5: **Evaluation on ML-Benchmark with different LLMs.** Left: completion rate. Right: comprehensive score.

**Ablation on different base LLMs.** Based on GPT-4o and GPT-4o-mini, Data Interpreter shows further improvement in task completion across a wide range of tasks, as illustrated in Figure 5. In machine learning tasks, LLMs like Qwen-72B-Chat (Bai et al., 2023) and Mixtral-8x7B (Jiang et al., 2024) performed comparably to GPT-3.5-Turbo, while smaller LLMs experienced performance degradation. Our Data Interpreter handled data loading and analysis effectively with smaller models but had limitations with tasks requiring advanced coding proficiency. Mixtral-8x7B achieved high completion rates in three tasks but faced challenges in the WSC task. Smaller LLMs also encountered execution failures due to restricted coding abilities when acquiring images or parsing webpage results, as shown in Figure 5.

## 5 CONCLUSION

In this work, we present the Data Interpreter, an LLM-based agent designed to tackle data science challenges via hierarchical graph representation. Our framework continuously monitors data changes and adapts to dynamic environments through iterative task refinement and graph optimization. It enhances data analysis and machine learning performance, and improves reasoning capabilities through hierarchical decomposition, fine-grained execution, validation, and iterative modifications. Combined with the LLM's planning and coding abilities, this approach effectively solves tasks requiring complex multi-step reasoning. Extensive evaluations demonstrate that our Data Interpreter outperforms various open-source frameworks in machine learning tasks, mathematical problems, and real-world applications, marking a significant advancement in the capabilities of LLM-based agents for data science.

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

## A  LIMITATIONS

**Insufficient diversity and complexity.** Our novel framework Data Interpreter outperforms other open-source frameworks on machine learning problems, yet are limited to entry-level Kaggle datasets and benchmarked against the capabilities of a junior human data scientist. These datasets are relatively small (under 500MB), with a limited number of columns (in the hundreds) and rows (in the tens of thousands), and mainly involve classification and regression tasks (as described in Appendix F.2). However, we have not yet evaluated our Data Interpreter on more challenging datasets involving large-scale data or complex tasks such as time series analysis, multi-label classification, or multi-table problems. In our future work, we plan to expand our dataset collection to include these types of problems to thoroughly evaluate our framework's performance and capabilities. **Precise self-improvement.** Human data scientists usually perform multiple experiments on a dataset, focusing on pipeline optimization and hyperparameter tuning Liu et al. (2021); Hutter et al. (2019). Our Data Interpreter integrates experience to enhance the node generation quality. The experience primarily involves tracking the progress of tasks and code. However, it does not use numerical feedback from multiple experiences to develop and refine specific strategies, such as increasing the learning rate or using an ensemble technique, to improve the performance continuously for a given dataset, thus lacking the capability for automatic self-improvement. In the future, we aim to address this limitation by developing mechanisms that allow our model to conduct multiple experiments and derive insights from the numerical feedback for a given dataset on its own. **DAG constraint detection mechanism.** Our current implementation does not include an explicit DAG constraint detection mechanism, we rely on the LLM's inherent ability to avoid cycles during task planning, as observed in our experiments. However, such mechanisms could enhance robustness in handling less structured domains or highly complex dependencies. Incorporating cycle detection and resolution strategies in future iterations would ensure improved reliability and adaptability across diverse applications. **Full-scale evaluation on mathematical problems.** For the MATH problem, our experiments are limited to level-5 problems, primarily due to the budget constraints, we will explore more cost-effective strategies for evaluating our Data Interpreter on a wider range of mathematical problems in future studies.

## B  BROADER IMPACT

Our work has the potential to significantly reduce the costs associated with a wide range of customized data science tasks, empowering professionals in the field to enhance their automation capabilities and efficiency. However, the flexibility of tools integration, while convenient for local code snippets integration, comes with potential risks. For instance, if users provide malicious code intended for unauthorized system penetration or web attacks, it could lead to security vulnerabilities. In our experiments, we mitigate this risk by prompting our Data Interpreter to check the codes before generating new codes. Additional saftguards against these risks include collaborating exclusively with LLMs that adhere to robust safety policies.

# C    IMPLEMTATION DETAILS

## C.1    PROGRAMMABLE NODE GENERATION

We illustrate the process of node generation process with tools.

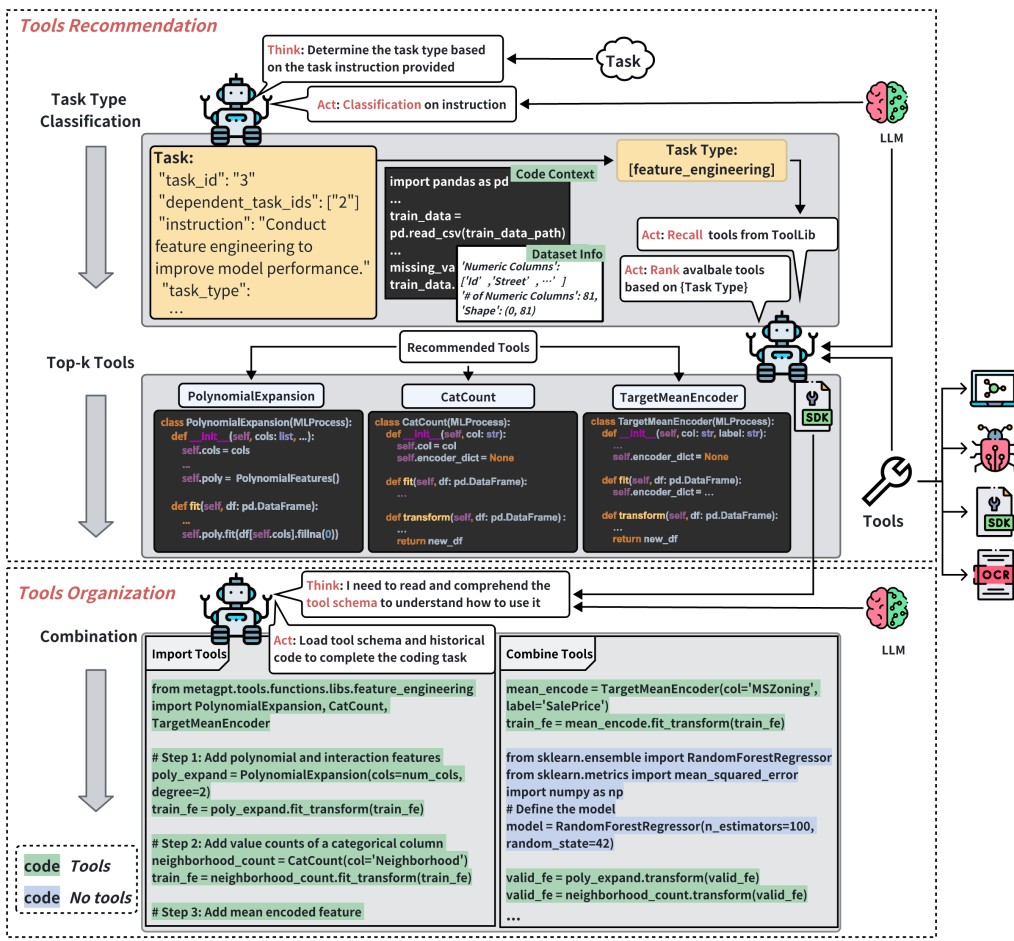

Figure 6: **Node generation pipeline in Data Interpreter.** Tools are initially selected based on task metadata classification, followed by tools organization process which combines multiple tools as necessary to accomplish the tasks.

### C.1.1    AN EXAMPLE OF TOOL SCHEMA

Below is an example of tool schema we design in our framework.

864
865
866
867
868
869
870
871
872
873
874
875
876
877
878
879
880
881
882
883
884
885
886
887
888
889
890
891
892
893
894
895
896
897
898
899
900
901
902
903
904
905
906
907
908
909
910
911
912
913
914
915
916
917

**Tool schema for a feature engineering tool**

```
type: class
description: Add value counts of a categorical column as new feature.
methods:
  __init__:
    type: function
    description: Initialize self.
    parameters:
      properties:
        col:
          type: str
          description: Column for value counts.
      required:
      - col
  fit:
    type: function
    description: Fit a model to be used in subsequent transform.
    parameters:
      properties:
        df:
          type: pd.DataFrame
          description: The input DataFrame.
      required:
      - df
  fit_transform:
    type: function
    description: Fit and transform the input DataFrame.
    parameters:
      properties:
        df:
          type: pd.DataFrame
          description: The input DataFrame.
      required:
      - df
    returns:
    - type: pd.DataFrame
      description: The transformed DataFrame.
  transform:
    type: function
    description: Transform the input DataFrame with the fitted model.
    parameters:
      properties:
        df:
          type: pd.DataFrame
          description: The input DataFrame.
      required:
      - df
    returns:
    - type: pd.DataFrame
      description: The transformed DataFrame.
```

### C.1.2   TOOLS DETAILS

The tools of our Data Interpreter are listed in Table 5

### C.1.3   TOOL USAGE PROMPTS

We use two types of prompts for tool utilization. For open-ended tasks, we use zero-shot prompts, and for machine-learning tasks, we use one-shot prompts as illustrated below.

Table 5: **Tools of our Data Interpreter.**

| Tool name | Tool type | Functions | Domain |
|---|---|---|---|
| FillMissingValue | Class | 4 | Machine learning |
| MinMaxScale | Class | 4 | Machine learning |
| StandardScale | Class | 4 | Machine learning |
| MaxAbsScale | Class | 4 | Machine learning |
| LabelEncode | Class | 4 | Machine learning |
| OneHotEncode | Class | 4 | Machine learning |
| OrdinalEncode | Class | 4 | Machine learning |
| RobustScale | Class | 4 | Machine learning |
| CatCount | Class | 4 | Machine learning |
| TargetMeanEncoder | Class | 4 | Machine learning |
| KFoldTargetMeanEncoder | Class | 4 | Machine learning |
| CatCross | Class | 5 | Machine learning |
| SplitBins | Class | 4 | Machine learning |
| GeneralSelection | Class | 4 | Machine learning |
| TreeBasedSelection | Class | 4 | Machine learning |
| VarianceBasedSelection | Class | 4 | Machine learning |
| PolynomialExpansion | Class | 4 | Machine learning |
| GPTvGenerator | Class | 3 | Multimodal |
| SDEngine | Class | 5 | Multimodal |
| scrape_web_playwright | Function | 1 | Common |

---

**Zero-shot tool usage prompt**

```
# Instruction
Write complete code for 'Current Task'. And avoid duplicating code from finished tasks
    , such as repeated import of packages, reading data, etc.
Specifically, {tool_type_usage_prompt}

# Capabilities
- You can utilize pre-defined tools in any code lines from 'Available Tools' in the
    form of Python Class.
- You can freely combine the use of any other public packages, like sklearn, numpy,
    pandas, etc..

# Available Tools (can be empty):
Each Class tool is described in JSON format. When you call a tool, import the tool
    first.
{tool_schemas}

# Constraints:
- Ensure the output new code is executable in the same Jupyter notebook with the
    previous tasks code has been executed.
- Always prioritize using pre-defined tools for the same functionality.
```

**One-shot tool usage prompt**

```
# Capabilities
- You can utilize pre-defined tools in any code lines from 'Available Tools' in the
    form of Python Class.
- You can freely combine the use of any other public packages, like sklearn, numpy,
    pandas, etc..

# Available Tools:
Each Class tool is described in JSON format. When you call a tool, import the tool
    from its path first.
{tool_schemas}

# Output Example:
when the current task is "do data preprocess, like fill missing value, handle outliers
    , etc.", the code can be like:
'''python
# Step 1: fill missing value
# Tools used: ['FillMissingValue']
from metagpt.tools.libs.data_preprocess import FillMissingValue

train_processed = train.copy()
test_processed = test.copy()
num_cols = train_processed.select_dtypes(include='number').columns.tolist()
if 'label' in num_cols:
    num_cols.remove('label')
fill_missing_value = FillMissingValue(features=num_cols, strategy='mean')
fill_missing_value.fit(train_processed)
train_processed = fill_missing_value.transform(train_processed)
test_processed = fill_missing_value.transform(test_processed)

# Step 2: handle outliers
for col in num_cols:
    low, high = train_processed[col].quantile([0.01, 0.99])
    train_processed[col] = train_processed[col].clip(low, high)
    test_processed[col] = test_processed[col].clip(low, high)
'''end

# Constraints:
- Ensure the output new code is executable in the same Jupyter notebook with the
    previous tasks code has been executed.
- Always prioritize using pre-defined tools for the same functionality.
- Always copy the DataFrame before processing it and use the copy to process.
```

# D EXPERIMENT DETAILS

## D.1 DATASET

**InfiAgent-DABench** InfiAgent-DABench focuses on evaluating the data analysis capabilities of agents. It comprises 257 data analysis problems, categorized into the following seven areas and their combinations: summary statistics, feature engineering, correlation analysis, machine learning, distribution analysis, outlier detection, and comprehensive data preprocessing. Each category includes problems of varying difficulty levels. Below, we present some specific prompt cases to provide an intuitive understanding of the task settings in InfiAgent-DABench.

**InfiAgent-DABench prompt**

```
1. category: ['Summary Statistics'] , level: easy.
 prompt: Please write a Python code snippet to Calculate the mean and standard
     deviation of the abs_diffsel column. based on the following details: The task is
     to { The mean and standard deviation should be calculated directly from the '
     abs_diffsel' column. Do not remove any outliers or modify the data prior to
     calculation. The mean and standard deviation should be computed directly from all
      available data points. }  and formatted as { @mean[mean_value] @std_dev[
     std_dev_value] where "mean_value" is a positive float number, rounded to two
     decimal places. where "std_dev_value" is a positive float number, rounded to two
     decimal places.. The data is stored in a file saved in "InfiAgent/examples/DA-
     Agent/data/da-dev-tables/ferret-Pitt-2-preinf-lib2-100_sitediffsel.csv", and the
     difficulty level is easy.
```

```
2. category: ['Feature Engineering', 'Correlation Analysis'] , level: medium.
prompt: Please write a Python code snippet to Create a new feature called 'FamilySize'
    by combining the 'SibSp' and 'Parch' columns, which represents the total number
    of family members a passenger had aboard the Titanic. Then, find the correlation
    coefficient between 'FamilySize' and 'Survived'. based on the following details:
    The task is to Create 'FamilySize' by adding up 'SibSp' and 'Parch', then
    calculate the Pearson correlation coefficient between 'FamilySize' and 'Survived'.
     and formatted as @correlation_coefficient[number] where "number" is the
    calculated Pearson correlation coefficient between 'FamilySize' and 'Survived',
    rounded to two decimal places.. The data is stored in a file saved in "InfiAgent/
    examples/DA-Agent/data/da-dev-tables/titanic.csv", and the difficulty level is
    medium.

3. category: ['Comprehensive Data Preprocessing', 'Distribution Analysis'] , level:
    hard.
 prompt: Please write a Python code snippet to 2. Preprocess the dataset by handling
    missing values in the "24-Hour Passes Purchased (midnight to 11:59 pm)" and "7-
    Day Passes Purchased (midnight to 11:59 pm)" columns. Use the mean imputation
    method to fill in the missing values. Then, analyze the distribution of the "
    Trips over the past 24-hours (midnight to 11:59pm)" column before and after the
    missing value imputation process. Evaluate if the imputation has significantly
    affected the distribution and what implications it has on the dataset analysis.
    based on the following details: The task is to Use the mean imputation method to
    fill in missing values for both the "24-Hour Passes Purchased (midnight to 11:59
    pm)" and "7-Day Passes Purchased (midnight to 11:59 pm)" columns. Then, calculate
     the mean, median, standard deviation, skewness, and kurtosis for the "Trips over
     the past 24-hours (midnight to 11:59pm)" column before and after imputation.
    and formatted as @pre_mean[mean_before] @pre_median[median_before] @pre_sd[
    sd_before] @pre_skewness[skew_before] @pre_kurtosis[kurt_before] @post_mean[
    mean_after] @post_median[median_after] @post_sd[sd_after] @post_skewness[
    skew_after] @post_kurtosis[kurt_after] where all variables represent the
    corresponding statistical values calculated before (prefix: pre) and after (
    prefix: post) the imputation, each rounded to two decimal places.. The data is
    stored in a file saved in "InfiAgent/examples/DA-Agent/data/da-dev-tables/2014_q4
    .csv", and the difficulty level is hard.
```

**ML-Benchmark** This dataset encompassed eight representative machine learning tasks categorized into three difficulty levels, ranging from easy (level 1) to most complex (level 3). Each task was accompanied by data, a concise description, standard user requirements, suggested steps, and metrics (see Table 13 in the Appendix). For tasks labeled as "toy", the data was not divided into training and test splits, which required the framework to perform data splitting during modeling.

**Open-ended task benchmark** To evaluate the ability to generalize to real-world tasks, we developed the Open-ended task benchmark, comprising 20 tasks. Each task required the framework to understand user needs, break down complex tasks, and execute code. They delineated their requirements, foundational data or sources, steps for completion, and specific metrics. The scope was broad, encompassing common needs like Optical Character Recognition (OCR), web search and crawling (WSC), automated email replies (ER), web page imitation (WPI), text-to-image conversion (T2I), image-to-HTML code generation (I2C), image background removal (IBR), and mini-game generation (MGG). We showcase about these tasks in Figure 11, Figure 13, and Figure 14 in the Appendix.

**MATH dataset** The MATH dataset Hendrycks et al. (2021) comprises 12,500 problems, with 5,000 designated as the test set, covering various subjects and difficulty levels. These subjects include Prealgebra (Prealg), Algebra, Number Theory (N.Theory), Counting and Probability (C.Prob), Geometry, Intermediate Algebra, and Precalculus (Precalc), with problems categorized from levels "1" to "5" based on difficulty. Following the setting of Wu et al. Wu et al. (2023c), we evaluated four typical problem types (C.Prob, N.Theory, Prealg, Precalc), excluding level-5 geometry problems from the test set.

## D.2 EVALUATION METRICS

In the MATH benchmark Hendrycks et al. (2021), accuracy served as the chosen evaluation metric, aligning with the setting proposed in Wu et al. (2023c); Hendrycks et al. (2021).

For the ML-Benchmark, three evaluation metrics were utilized: completion rate (CR), normalized performance score (NPS), and comprehensive score (CS). These metrics provided comprehensive insights into the model's performance and were defined as follows:

***Completion rate (CR)***: In the task requirements description, there were $T$ steps, and the task completion status of each step was denoted by a score $s_t$, with a maximum score $s_{max}$ of 2 and a minimum score $s_{min}$ of 0. The task completion status categories were defined as follows: missing (score of 0), fail (score of 0), success - non-compliant (score of 1), success-compliant (score of 2), and optional step (not involved in scoring). To measure the completion level, we proposed a completion ratio where the numerator was the sum of scores $s_t$ for each step, and the denominator was the sum of the maximum possible scores for all steps ($s_{max} \times T$):

$$\text{CR} = \frac{\sum_{t=1}^{T} s_t}{s_{max} \times T}. \tag{3}$$

***Normalized performance score (NPS)***: In our ML-Benchmark, each task was associated with its evaluation metric, which may vary between tasks, including metrics such as accuracy, F1, AUC and RMSLE, etc. For metrics such as accuracy, F1, and AUC, we presented the raw values to facilitate comparison across identical data tasks. We normalize all performance values $s$:

$$\text{NPS} = \begin{cases} \dfrac{1}{1+s}, & \text{if } s \text{ is smaller the better} \\ s, & \text{otherwise.} \end{cases} \tag{4}$$

This transformation ensured that loss-based metrics like RMSLE are scaled from 0 to 1, with higher normalized performance score values indicating better performance.

***Comprehensive score (CS)***: To simultaneously assess both the completion rate of task requirements and the performance of generated machine learning models, we calculated the weighted sum of CR and NPS as follows:

$$\text{CS} = 0.5 \times \text{CR} + 0.5 \times \text{NPS}. \tag{5}$$

Considering the lack of unified performance standards for open-ended tasks, we default to NPS $= 0$ and directly equate CS to CR.

### D.3 ADDITIONAL RESULTS

#### D.3.1 ADDITIONAL RESULTS OF ML-BENCHMARK AND MATH DATASET

For a deeper understanding, Table 6 presents the results on the ML-benchmark for both Completion Rate and Normalized Performance Score metrics. Additionally, Table 12 showcases the results of ablation experiments on the ML-benchmark, focusing on the completion rate (CR) and normalized performance score (NPS).

Table 6: **Additional performance comparisons on ML benchmark.** "WR", "BCW", "ICR", "SCTP", and "SVPC" represent "Wine recognition"", "Breast cancer wisconsin", "ICR - Identifying age-related conditions", "Santander customer transaction prediction", and "Santander value prediction challenge", respectively. "Avg." denotes "Average".

| Model / Task | WR | BCW | Titanic | House Prices | SCTP | ICR | SVPC | Avg. |
|---|---|---|---|---|---|---|---|---|
| *Completion rate* | | | | | | | | |
| AutoGen | 0.92 | **1.00** | 0.92 | 0.83 | 0.83 | 0.83 | 0.83 | 0.88 |
| OpenInterpreter | **1.00** | 0.90 | 0.92 | 0.88 | 0.85 | 0.91 | 0.88 | 0.90 |
| TaskWeaver | **1.00** | **1.00** | 0.83 | 0.88 | 0.67 | 0.83 | 0.80 | 0.86 |
| XAgent | **1.00** | **1.00** | 0.83 | 0.83 | 0 | 0.67 | 0 | 0.62 |
| OpenDevin | **1.00** | **1.00** | 0.92 | 1.00 | 1.00 | 0.83 | 1.00 | 0.96 |
| **Data Interpreter** | **1.00** | **1.00** | **1.00** | **1.00** | **1.00** | **1.00** | **1.00** | **1.00** |
| *Normalized performance score* | | | | | | | | |
| AutoGen | **1.00** | 0.97 | 0.82 | 0.88 | 0.82 | 0.71 | 0.63 | 0.83 |
| OpenInterpreter | **1.00** | 0.96 | 0.81 | 0.87 | 0.52 | 0.25 | 0 | 0.63 |
| TaskWeaver | **1.00** | 0.96 | 0.43 | 0.49 | 0 | 0.65 | 0.17 | 0.53 |
| XAgent | **1.00** | 0.94 | 0 | 0 | 0 | 0 | 0 | 0.28 |
| OpenDevin | **0.96** | 0.96 | 0.81 | 0.87 | 0.86 | 0.62 | 0.45 | 0.79 |
| **Data Interpreter** | 0.96 | **0.99** | **0.82** | **0.91** | **0.89** | **0.91** | **0.77** | **0.89** |

Table 7: **Additional performance comparisons on MATH dataset.** "Avg." and "Std." denotes "Average", "Standard Deviation" respectively.

| Category | MathChat | AutoGen | Data Interpreter | | | | |
|---|---|---|---|---|---|---|---|
| | | | Avg. | Trial1 | Trail2 | Trail3 | Std.(%) |
| C.Prob | 0.52 | 0.59 | 0.68 | 0.70 | 0.66 | 0.68 | 2.05 |
| N.Theory | 0.60 | 0.66 | 0.82 | 0.81 | 0.82 | 0.82 | 0.99 |
| Prealg | 0.60 | 0.63 | 0.74 | 0.73 | 0.75 | 0.75 | 1.20 |
| Precalc | 0.19 | 0.12 | 0.29 | 0.28 | 0.30 | 0.29 | 1.13 |

## D.4 OVERHEAD ANALYSIS

We compared our token cost (average per task) and inference time (average per task) across the ML-Benchmark, Open-ended Task Benchmark, MATH Dataset, and InfriAgent-DABench, while also reporting our performance. Our framework demonstrates a state-of-the-art performance with competitive efficiency.

Table 8: **Overhead analysis on MATH Dataset.** "Cost" represents the total cost in USD, "Time" indicates the total execution time in seconds, "Avg." denotes "Average".

| Model / Metric | Cost ($)↓ | Time (s)↓ | Accuracy↑ |
|---|---|---|---|
| AutoGen | **0.242** | **120.99** | 0.500 |
| **Data Interpreter** | 0.336 | 211.57 | **0.633** |

Table 9: **Overhead analysis on InfriAgent-DABench.** "Cost" represents the total cost in USD, "Time" indicates the total execution time in seconds, "Avg." denotes "Average".

| Model / Metric | Cost ($)↓ | Time (s)↓ | Accuracy↑ |
|---|---|---|---|
| AutoGen (GPT-4o) | 0.112 | **42.42** | 88.72 |
| AutoGen (GPT-4-0613) | 0.423 | 45.69 | 71.49 |
| **Data Interpreter** (GPT-4o) | **0.017** | 49.44 | **94.93** |
| **Data Interpreter** (GPT-4-0613) | 0.311 | 51.09 | 73.55 |

On specific domains like MATH Dataset (See Table 8) and InfriAgent-DABench (See Table 9), Data Interpreter consistently shows superior accuracy (63.3% and 94.93% respectively) while maintaining competitive efficiency, as demonstrated in Table 8 and Table 9. Notably, on InfriAgent-DABench, our approach achieves better performance with lower cost (0.017 USD vs. 0.112 USD) compared to AutoGen.

On ML-Benchmark (See Table 10), Data Interpreter achieves the highest comprehensive score (0.95) among all frameworks, though with moderate cost (0.84 USD) and inference time (237.31s), as shown in table 10. While frameworks like OpenInterpreter achieve lower costs (0.21 USD) through one-time code generation, they show inferior performance (0.77).

In Table 11, for open-ended tasks, Data Interpreter significantly outperforms baselines with a comprehensive score of 0.953, maintaining reasonable cost (0.34 USD) compared to OpenDevin (1.41 USD) and AutoGen (0.30 USD).

Table 10: **Overhead analysis on ML Benchmark.** "SCTP", and "SVPC" represent "ICR - Identifying age-related conditions", "Santander customer transaction prediction", and "Santander value prediction challenge", respectively. "Cost" represents the total cost in USD, "Time" indicates the total execution time in seconds, "Avg." denotes "Average".

| Model / Task | Titanic | House | ICR | SCTP | SVPC | Avg. |
|---|---|---|---|---|---|---|
| *Cost ($)↓* | | | | | | |
| AutoGen | 0.08 | 0.25 | 0.19 | 0.48 | 0.58 | **0.32** |
| OpenInterpreter | **0.26** | **0.15** | 0.27 | **0.18** | **0.21** | **0.21** |
| OpenDevin | 2.66 | 3.01 | 3.35 | 3.24 | 2.78 | 3.01 |
| TaskWeaver | 0.35 | 0.38 | 0.36 | 0.29 | 0.48 | 0.37 |
| XAgent | 21.15 | 17.16 | 27.81 | 14.12 | 20.23 | 20.09 |
| **Data Interpreter** | 0.65 | 0.84 | **0.76** | 0.54 | 1.41 | 0.84 |
| *Time (s)↓* | | | | | | |
| AutoGen | **124.71** | **84.11** | **136.91** | 280.60 | **244.04** | **174.07** |
| OpenInterpreter | 116.66 | 132.00 | 170.00 | 239.00 | 296.00 | 190.73 |
| OpenDevin | 164.00 | 133.00 | 148.00 | 282.00 | 212.00 | 187.80 |
| TaskWeaver | 109.76 | 279.25 | 151.97 | **182.13** | 119.62 | 168.55 |
| XAgent | 5400.00 | 5107.00 | 5400.00 | 6023.00 | 9000.00 | 6186.00 |
| **Data Interpreter** | 168.01 | 193.21 | 184.77 | 244.39 | 396.17 | 237.31 |
| *Comprehensive Score↑* | | | | | | |
| AutoGen | 0.87 | 0.86 | 0.83 | 0.77 | 0.73 | 0.86 |
| OpenInterpreter | 0.86 | 0.87 | 0.68 | 0.58 | 0.44 | 0.77 |
| OpenDevin | 0.87 | **0.94** | **0.93** | 0.73 | 0.73 | 0.88 |
| TaskWeaver | 0.63 | 0.68 | 0.34 | 0.74 | 0.48 | 0.69 |
| XAgent | 0.42 | 0.42 | 0.00 | 0.34 | 0.01 | 0.45 |
| **Data Interpreter** | **0.91** | 0.96 | 0.94 | **0.96** | **0.89** | **0.95** |

Table 11: **Overhead comparison on Open-ended Tasks.** "OCR", "WSC", "WPI", and "IBR" represent "Optical Character Recognition", "Web Search and Crawling", "Web Page Imitation", and "Image Background Removal", respectively. "Cost" represents the total cost in USD, "Time" indicates the total execution time in seconds, "Avg." denotes "Average".

| Model / Task | OCR | WSC | WPI | IBR | Avg. |
|---|---|---|---|---|---|
| *Cost ($)↓* | | | | | |
| AutoGen | 0.10 | 0.18 | 0.43 | 0.48 | 0.30 |
| OpenInterpreter | **0.28** | **0.08** | **0.15** | **0.07** | **0.15** |
| OpenDevin | 1.27 | 1.88 | 1.26 | 1.24 | 1.41 |
| **Data Interpreter** | 0.275 | 0.69 | 0.23 | 0.18 | 0.34 |
| *Time (s)↓* | | | | | |
| AutoGen | **68.85** | 57.28 | 154.46 | 79.26 | 90.05 |
| OpenInterpreter | 133.00 | 109.00 | 102.00 | 68.00 | **103.00** |
| OpenDevin | 190.00 | 196.00 | 94.00 | 146.00 | 156.50 |
| **Data Interpreter** | 77.00 | **293.00** | 65.00 | 34.00 | 117.25 |
| *Comprehensive Score↑* | | | | | |
| AutoGen | 0.67 | 0.65 | 0.26 | **1.00** | 0.65 |
| OpenInterpreter | 0.50 | 0.30 | 0.36 | **1.00** | 0.54 |
| OpenDevin | 0.60 | 0.87 | 0.16 | **1.00** | 0.66 |
| **Data Interpreter** | **0.85** | **0.96** | **1.00** | **1.00** | **0.95** |

### D.4.1 ABLATION STUDY

Here we provide detailed ablation study results on core modules.

Table 12: **Ablation on core modules.** Evaluated with CR, NPS and CS on ML-Benchmark. "IGR" stands for Iterative Graph Refinement, and "PNG" denotes Programmable Node Generation. "ICR", "SCTP", and "SVPC" represent "ICR - Identifying age-related conditions", "Santander customer transaction prediction", and "Santander value prediction challenge", respectively.

| Code execution | IGR | PNG | House Prices | SCTP | SVPC | ICR | Avg. |
|---|---|---|---|---|---|---|---|
| *Completion rate* | | | | | | | |
| ✓ | | | 0.58 | 0.33 | 0.67 | 0.33 | 0.48 |
| ✓ | ✓ | | 1.00 | 1.00 | 0.92 | 0.88 | 0.95 |
| ✓ | ✓ | ✓ | **1.00** | **1.00** | **1.00** | **1.00** | **1.00** |
| *Normalized performance score* | | | | | | | |
| ✓ | | | 0.43 | 0 | 0.64 | 0 | 0.27 |
| ✓ | ✓ | | 0.91 | 0.82 | 0.68 | 0.60 | 0.75 |
| ✓ | ✓ | ✓ | **0.91** | **0.89** | **0.77** | **0.91** | **0.87** |
| *Comprehensive score* | | | | | | | |
| ✓ | | | 0.51 | 0.17 | 0.66 | 0.17 | 0.37 |
| ✓ | ✓ | | 0.96 | 0.91 | 0.80 | 0.74 | 0.85 |
| ✓ | ✓ | ✓ | **0.96** | **0.95** | **0.89** | **0.96** | **0.94** |

# E  ADDITIONAL EXAMPLES

## E.1  AN EXAMPLE OF TASK GRAPH

Here is the prompt used to generate the task graph.

---
**Prompt for task graph generator**

```
PLAN_PROMPT = """
# Context:
{context}
# Available Task Types:
{task_type_desc}
# Task:
Based on the context, write a plan or modify an existing plan of what you should do to
    achieve the goal. A plan consists of one to {max_tasks} tasks.
If you are modifying an existing plan, carefully follow the instruction, don't make
    unnecessary changes. Give the whole plan unless instructed to modify only one task
    of the plan.
If you encounter errors on the current task, revise and output the current single task
    only.
Output a list of jsons following the format:
[
    {{
        "task_id": str = "unique identifier for a task in plan, can be an ordinal",
        "dependent_task_ids": list[str] = "ids of tasks prerequisite to this task",
        "instruction": "what you should do in this task, one short phrase or sentence
            ",
        "task_type": "type of this task, should be one of Available Task Types",
    }},
    ...
]
"""
```
---

Here is an example of a task graph. The user requirement is: "This is a dataset featuring sensor readings from industrial machines, aimed at predicting machine operational status (normal or faulty). Visualize the analysis and prediction results with high-quality graphs. Train data path: {train_path}, eval data path: {eval_path}."

---
**Task graph example**

```
[
    {
        "task_id": "1",
        "dependent_task_ids": [],
        "instruction": "Perform data loading and preliminary exploration of the train
            and eval datasets. Fill missing values and apply MinMax scaling.",
        "task_type": "eda"
    },
    {
        "task_id": "2",
        "dependent_task_ids": [
            "1"
        ],
        "instruction": "Conduct correlation analysis and provide descriptive
            statistics.",
        "task_type": "eda"
    },
    {
        "task_id": "3",
        "dependent_task_ids": [
            "1"
        ],
        "instruction": "Perform outlier detection using Isolation Forest to identify
            and handle anomalies.",
        "task_type": "eda"
    },
    {
        "task_id": "4",
        "dependent_task_ids": [
            "2",
```
---

```
                "3"
            ],
            "instruction": "Execute feature engineering, including General Selection,
                Target Mean Encoding, and Variance Based Selection to prepare features for
                model training.",
            "task_type": "feature_engineering"
        },
        {
            "task_id": "5",
            "dependent_task_ids": [
                "4"
            ],
            "instruction": "Split the data and train predictive models using Random Forest
                and XGBoost.",
            "task_type": "model_train"
        },
        {
            "task_id": "6",
            "dependent_task_ids": [
                "5"
            ],
            "instruction": "Evaluate the model's performance and generate an evaluation
                report.",
            "task_type": "model_evaluate"
        },
        {
            "task_id": "7",
            "dependent_task_ids": [
                "5",
                "6"
            ],
            "instruction": "Visualize the analysis and prediction results, including
                classification reports and confusion matrix, and serialize the model.",
            "task_type": "visualization"
        }
    ]
```

## E.2 PROMPTS FOR ACTION GRAPH

Data Interpreter utilizes LLMs to generate an action graph for each task. For each task node, we maintain execution context and task graph state via plan status, and generate executable code using the following prompt:

**Prompt for action graph generator**

```
PLAN_STATUS = """
## Finished Tasks
### code
```python
{code_written}
```

### execution result
{task_results}

## Current Task
{current_task}

## Task Guidance
Write complete code for 'Current Task'. And avoid duplicating code from 'Finished
    Tasks', such as repeated import of packages, reading data, etc.
Specifically, {guidance}
"""

Action_Graph_Prompt = """
# User Requirement
{project_requirement}

# Plan Status
{plan_status}

# Tool Info
```

```
{tool_info}

# Constraints
- Take on Current Task if it is in Plan Status, otherwise, tackle User Requirement
    directly.
- Ensure the output new code is executable in the same Jupyter notebook as the
    previous executed code.
- Always prioritize using pre-defined tools for the same functionality.

# Output
While some concise thoughts are helpful, code is absolutely required. Always output
    one and only one code block in your response. Output code in the following format:
```python
your code
```
"""
```

### E.3 EXAMPLE OF DYNAMIC TASK GRAPH REFINEMENT

This section details how Data Interpreter resolves task failures and refines the task graph dynamically. Initially, the task graph is created as described in Appendix E.1. When encountering task execution failures (e.g., Task 4: feature engineering), Data Interpreter utilizes a reflection-based debugging prompt (REFLECTION_PROMPT) to iteratively analyze errors and propose improved implementations.

**Prompt for reflection and debugging**

```
REFLECTION_PROMPT = """
[example]
Here is an example of debugging with reflection.
{debug_example}
[/example]

[context]
{context}

[previous impl]:
{previous_impl}

[instruction]
Analyze your previous code and error in [context] step by step, provide me with
    improved method and code. Remember to follow [context] requirement. Don't forget
    to write code for steps behind the error step.
Output a json following the format:
```json
{{
    "reflection": str = "Reflection on previous implementation",
    "improved_impl": str = "Refined code after reflection.",
}}
```
"""
```

After repeated failures (e.g., three unsuccessful attempts at executing the action graph), Data Interpreter restructures the task graph: Tasks 1-3 remain unchanged, but Task 4 is simplified to basic feature creation, a new Task 5 for feature selection is introduced, and subsequent tasks (e.g., original Task 5 becoming Task 6) are automatically reindexed with updated dependencies, as shown below:

**Example of refined task graph**

```
...
{
    "task_id": "4",
    "dependent_task_ids": [
            "2",
            "3"
        ],
    "instruction": "Create engineered features from sensor readings",
    "task_type": "feature_engineering"
},
{
    "task_id": "5",
    "dependent_task_ids": [
            "4",
        ],
    "instruction": "Perform feature selection using statistical methods and importance
        analysis",
    "task_type": "feature_engineering"
},
 {
        "task_id": "6",
        "dependent_task_ids": [
            "4",
            "5"
        ],
        "instruction": "Train a predictive model to determine machine status",
        "task_type": "model_train"
    },
    ...
```

### E.4 RUNTIME RESULTS OF TASK GRAPH

We provide three distinct runtime results of our model, Data Interpreter, to offer an in-depth demonstration of its capabilities. These results meticulously showcase the intricacies of the task graph, action graph, and the overall graph structure as shown in Figure 7.

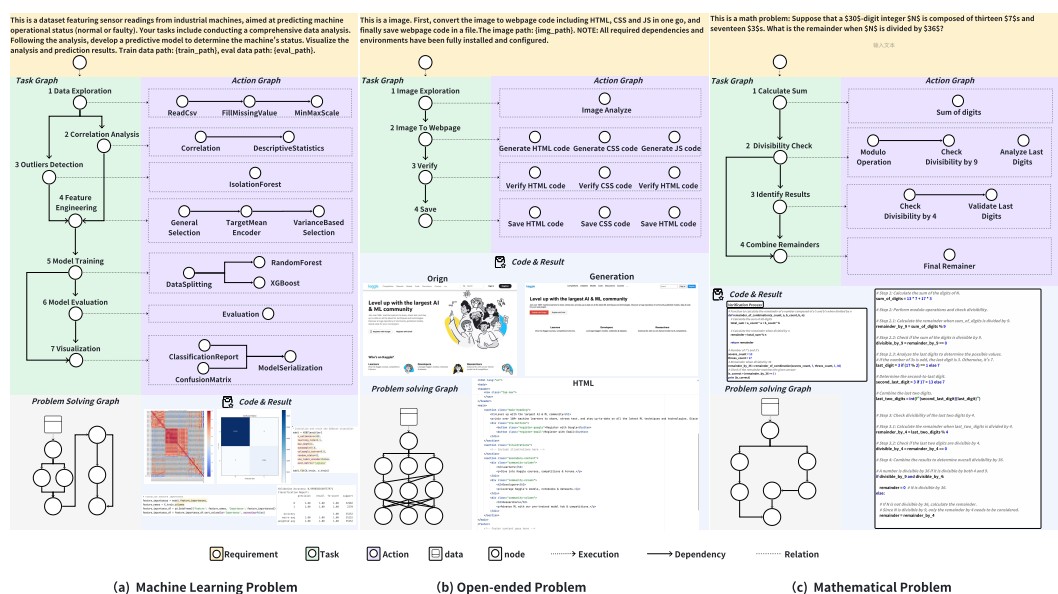

Figure 7: **Runtime examples of Data Interpreter**: machine learning, webpage imitation, and math problem solving

## E.5   Additional results of Open-ended tasks

We present the results by the Data Interpreter of several open-ended tasks in two figures: tasks 8, 9, 10, and 13 in Figure 8, and tasks 4, 14, and 15 in Figure 9.

## E.6   Result of data visualization

Figure 10 illustrates the results of data analysis and visualization of the Data Interpreter.

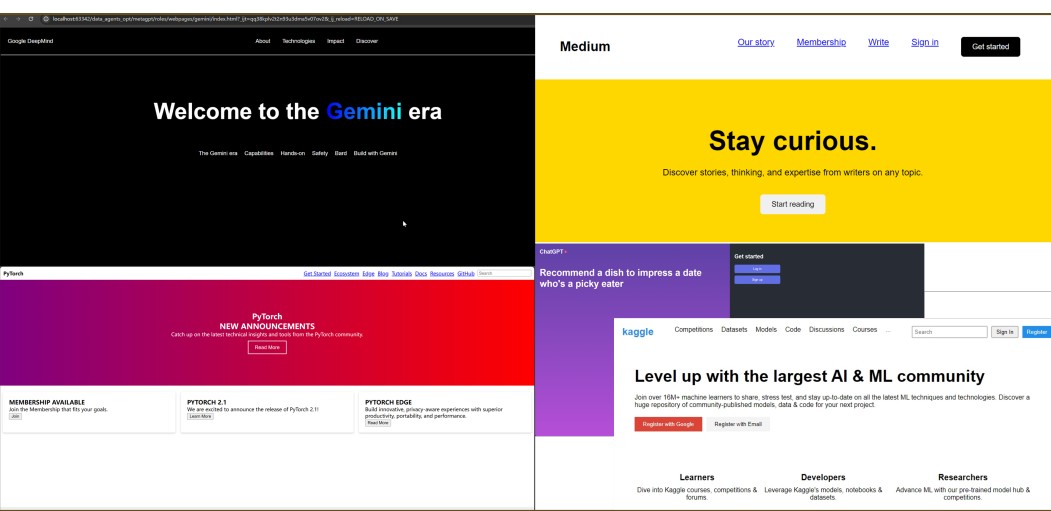

Figure 8: Web page imitation by Data Interpreter

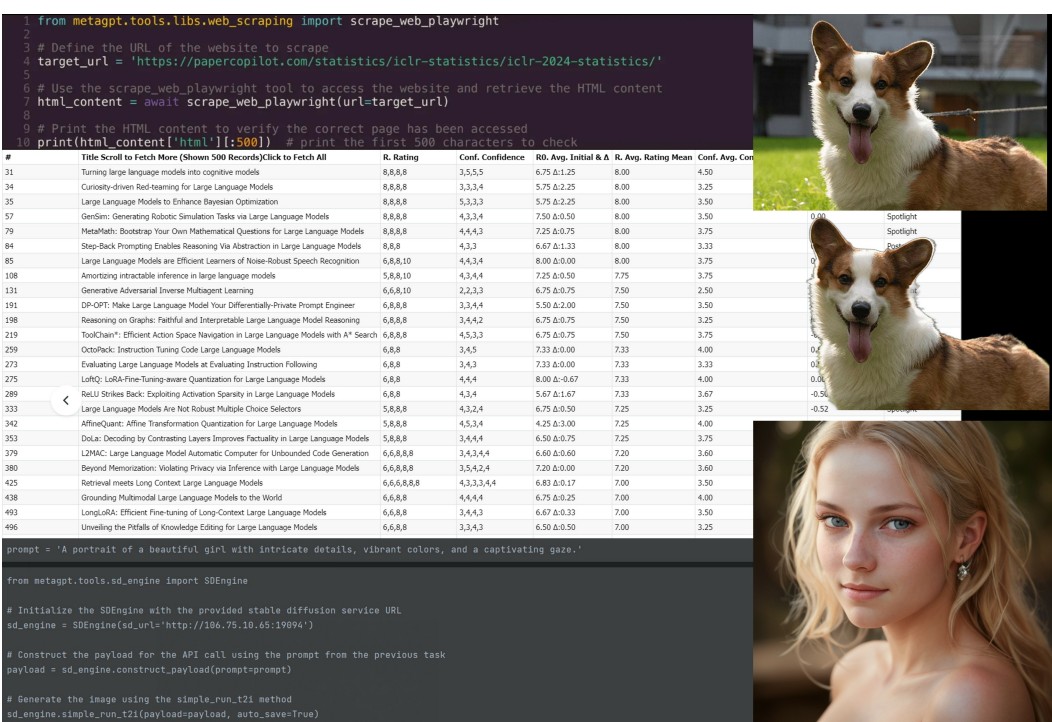

Figure 9: Image background removal / text-to-image / web search and crawling by Data Interpreter

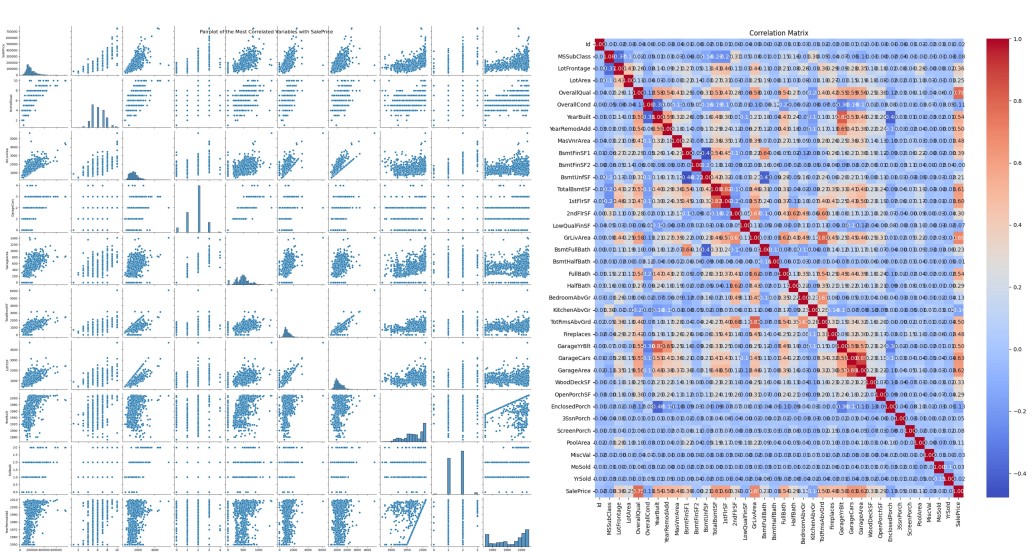

Figure 10: Data analysis and visualization capabilities of Data Interpreter

# F DETAILS OF DATASETS

## F.1 OPEN-ENDED TASK DETAILS

Figures 11 to 14 showcase several typical open-ended tasks in the following illustrations. For each task, we include the necessary data, user requirements, and assessment pipeline.

## F.2 ML-BENCHMARK DATASET DESCRIPTION

Here are the details about the ML-Benchmark dataset. We collect several typical datasets from Kaggle[1] and machine learning. Details are in Table 13

---

**(1) OCR (Task 1-3)**

**Scenario Description:** Scan all the necessary fields and amounts from the given file and then create an Excel sheet with the extracted data

**User Requirement:** This is an English invoice image.
Your goal is to perform OCR on the image, extract the total amount from ocr result and save as table, using PaddleOCR.
The PaddleOCR environment has been fully installed, try to use Paddleocr as much as possible.
Image path: ./workspace/CORD_test/image/receipt_00001.png

**Pipeline Requirement:**
1.Load and read images from a given folder/path
2.Install OCR tools/software
3.Using OCR tools/software to extract necessary fields and amounts
4.Collect results and convert them to a DataFrame
5.Save the result in a csv/xlsx forma

**Performance Requirement:** Recall / Precision / Accuracy

**Data:**
- Task 1:                    - Task 2:                    - Task 3:

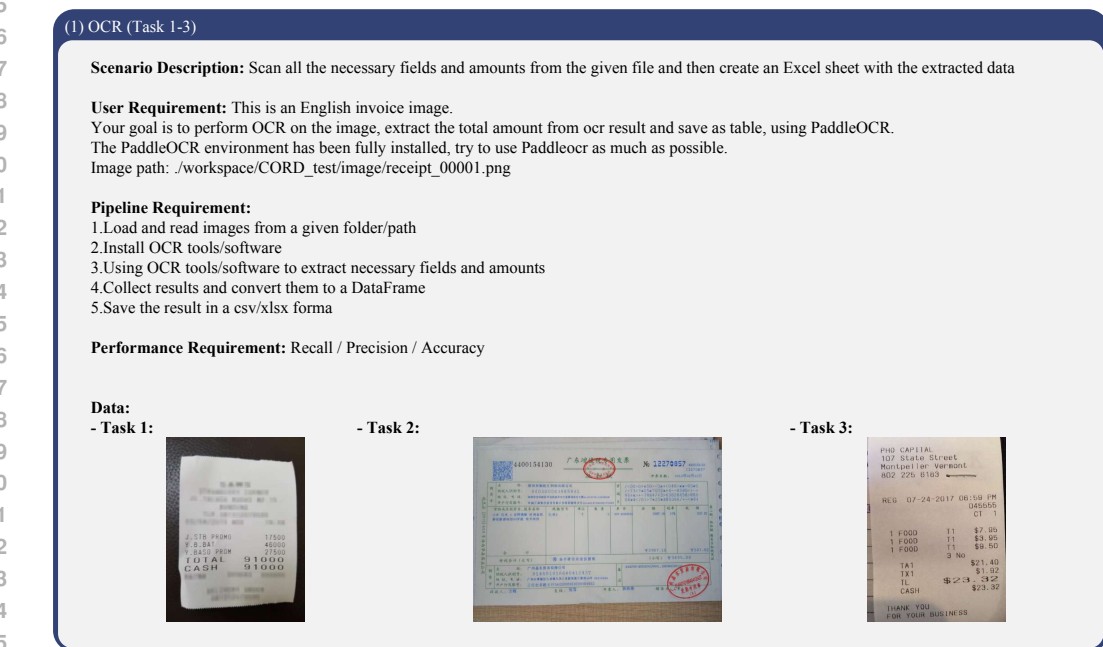

---

**(2) Web search and crawling (Task 4-7)**

**Scenario Description:** Crawling and organizing web form information

**Data: -**

**Pipeline Requirement:**
1.Open target URL
2.Select and filter the required information
3.Download or transform the data, convert them into a specified format
4.Output in a tabular form

**Performance Requirement:** Recall / Precision / Accuracy

**User Requirement:**
- Task 4:
Get data from `paperlist` table in https://papercopilot.com/statistics/iclr-statistics/iclr-2024-statistics/, and save it to a csv file. paper title must include `multiagent` or `large language model`.
notice: print key variables

---

Figure 11: **Open-ended task cases (OCR and web search and crawling)** We present task 4, omitting similar tasks for brevity.

---

[1]https://www.kaggle.com/

---

**(3) Email reply  (Task 8)**

**Scenario Description:** Filter through my emails and respond to them as necessary

**User Requirement:** You are an agent that automatically reads and replies to emails. I will give you your Outlook email account and password. You need to check the content of the latest email and return it to me. If the email address suffix of this email is @communication.microsoft.com, please automatically reply with "I've received your email and will reply as soon as possible. Thank you!"
Email account: englishgpt@outlook.com
Email Password: xxxx

**Data: -**

**Pipeline Requirement:**
1. Login to the target email account
2. Summarize and filter the email content accordingly.
3. set up an automatic reply to the sender with an email address that ends with a specific domain name.

**Performance Requirement: -**

---

**(4) Web page imitation (Task 9-13)**

**Scenario Description:** Using Selenium and WebDriver to access a webpage and convert it to an image, with the assistance of GPT-4V to mimic the creation of a one-page website.

**- Task 10:**
This is a URL of webpage: https://pytorch.org/. Firstly, utilize Selenium and WebDriver for rendering. Secondly, convert image to a webpage including HTML, CSS and JS in one go. Finally, save webpage in a file.
NOTE: All required dependencies and environments have been fully installed and configured.

**- Task 11:**
This is a URL of webpage: https://www.kaggle.com/. Firstly, utilize Selenium and WebDriver to render the webpage, ensuring the browser window is maximized for an optimal viewing experience. Secondly, convert image to a webpage including HTML, CSS and JS in one go. Finally, save webpage in a file. NOTE: All required dependencies and environments have been fully installed and configured.

**- Task 12:**
This is a URL of webpage: https://chat.openai.com/auth/login. Firstly, utilize Selenium and WebDriver to render the webpage, ensuring the browser window is maximized for an optimal viewing experience. Secondly, convert image to a webpage including HTML, CSS and JS in one go. Finally, save webpage in a file. NOTE: All required dependencies and environments have been fully installed and configured.

**Data:  (Task 10-12 in order)**

**Pipeline Requirement:**
1. Open a target Web URL
2. Transform the Website into an image
3. Send the image to GPT-4V via API
4. Request a similar website generation using the code.

**Performance Requirement:** Similarity/Correctness

---

Figure 12: **Open-ended task cases (email reply and web page imitation).** We present tasks 10-12, omitting similar tasks for brevity.

**(5) Image Background Removal (Task 14)**

**Scenario Description:** Remove the background of a given image

**User Requirement:** This is an image, you need to use python toolkit rembg remove the background of the image. image path:'./data/lxt.jpg'; save path:'./data/lxt_result.jpg'

**Data:**

**Pipeline Requirement:**
1. Read a local image
2. Install image background removal tools/software
3. Using background removal tools/software to remove the background of the target image
4. Save the new image

**Performance Requirement:** Correctness

---

**(6) Text2Img  (Task 15)**

**Scenario Description:** Use SD tools to generate images

**User Requirement:** I want to generate an image of a beautiful girl using the stable diffusion text2image tool, sd_url=""

**Data: -**

**Pipeline Requirement: -**

**Performance Requirement: -**

---

**(7) Image2Code (Task 16-17)**

**Scenario Description:** Web code generation

**User Requirement:**
**- Task 16:**
This is a image. First, check if the path exists, then convert the image to webpage code including HTML, CSS and JS in one go, and finally save webpage code in a file.The image path: ./medium.png .NOTE: All required dependencies and environments have been fully installed and configured.

**- Task 17:**
This is a image. First, check if the path exists, then convert the image to webpage code including HTML, CSS and JS in one go, and finally save webpage code in a file.The image path: ./gemini.png .NOTE: All required dependencies and environments have been fully installed and configured.

**Data:   (Task 16-17 in order)**

**Pipeline Requirement: -**

**Performance Requirement: -**

Figure 13: **Open-ended task cases (image background removal, text-to-image, and image-to-code)**

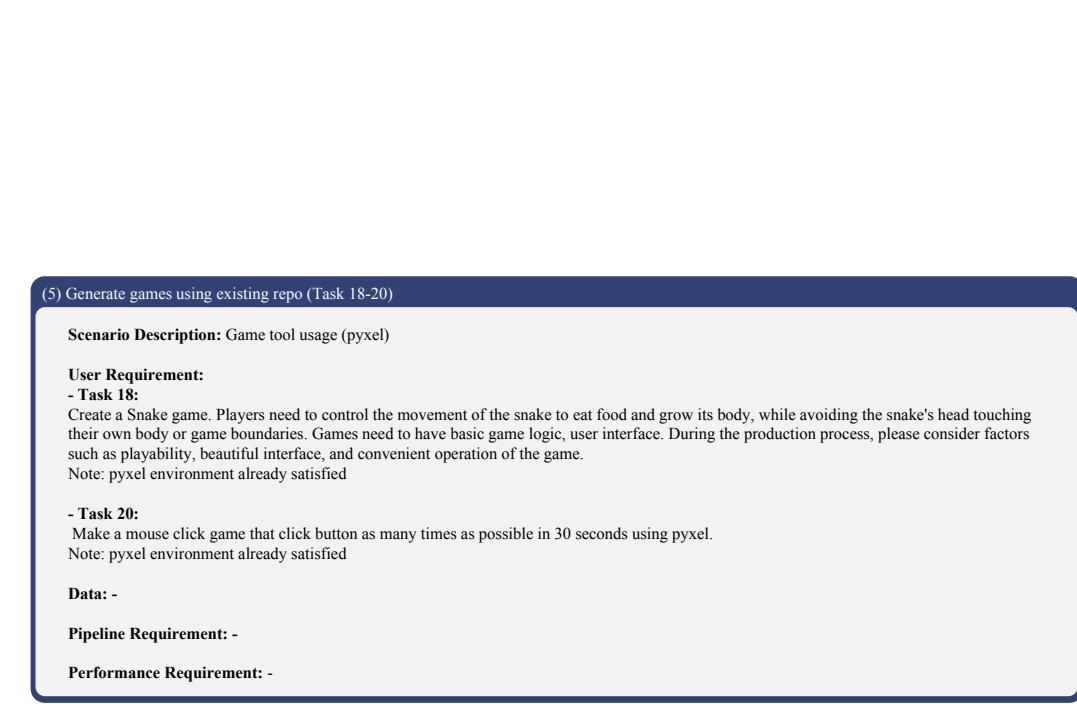

Figure 14: **Open-ended task cases (mini-game generation)** We present tasks 18 and 20, omitting similar tasks for brevity.

Table 13: Details of the ML-Benchmark dataset, including dataset name, description, standard user requirements, dataset type, task type, difficulty, and metric used.

| ID | Dataset Name | User Req. | Dataset Type | Dataset Description | Task Type | Difficulty | Metric |
|---|---|---|---|---|---|---|---|
| 01 | Iris | Run data analysis on sklearn Iris dataset, including a plot | Toy | Suitable for EDA, simple classification and regression | EDA | 1 | |
| 02 | Wine recognition | Run data analysis on sklearn Wine recognition dataset, include a plot, and train a model to predict wine class with 20% as test set, and show prediction accuracy | Toy | Suitable for EDA, simple classification and regression | Classification | 1 | ACC |
| 03 | Breast Cancer | Run data analysis on sklearn Wisconsin Breast Cancer dataset, include a plot, train a model to predict targets (20% as validation), and show validation accuracy | Toy | Suitable for EDA, binary classification to predict benign or malignant | Classification | 1 | ACC |
| 04 | Titanic | This is a Titanic passenger survival dataset, and your goal is to predict passenger survival outcomes. The target column is Survived. Perform data analysis, data pre-processing, feature engineering, and modeling to predict the target. Report accuracy on the eval data. Train data path: 'dataset\titanic\split_train.csv', eval data path: 'dataset\titanic\split_eval.csv'. | Beginner | Binary classification of survival, single table | Classification | 2 | ACC |
| 05 | House Prices | This is a house price dataset, and your goal is to predict the sale price of a property based on its features. The target column is SalePrice. Perform data analysis, data pre-processing, feature engineering, and modeling to predict the target. Report RMSE between the logarithm of the predicted value and the logarithm of the observed sales price on the eval data. Train data path: 'dataset\house-prices-advanced-regression-techniques\split_train.csv', eval data path: 'dataset\house-prices-advanced-regression-techniques\split_eval.csv'. | Beginner | Predicting house prices through property attributes, regression, single table | Regression | 2 | RMSLE |
| 06 | Santander Customer | This is a customer's financial dataset. Your goal is to predict which customers will make a specific transaction in the future. The target column is the target. Perform data analysis, data preprocessing, feature engineering, and modeling to predict the target. Report AUC on the eval data. Train data path: 'dataset\santander-customer-transaction-prediction\split_train.csv', eval data path: 'dataset\santander-customer-transaction-prediction\split_eval.csv'. | Industry | Binary classification to predict customer transactions, single table | Classification | 2 | AUC |
| 07 | ICR - Identifying | This is a medical dataset with over fifty anonymized health characteristics linked to three age-related conditions. Your goal is to predict whether a subject has or has not been diagnosed with one of these conditions. The target column is Class. Perform data analysis, data preprocessing, feature engineering, and modeling to predict the target. Report F1 Score on the eval data. Train data path: 'dataset\icr-identify-age-related-conditions\split_train.csv', eval data path: 'dataset\icr-identify-age-related-conditions\split_eval.csv'. | Industry | Binary classification of health symptoms, single table | Classification | 2 | F1 |
| 08 | Santander Value | This is a customer's financial dataset. Your goal is to predict the value of transactions for each potential customer. The target column is the target. Perform data analysis, data preprocessing, feature engineering, and modeling to predict the target. Report RMSLE on the eval data. Train data path: 'dataset\santander-value-prediction-challenge\split_train.csv', eval data path: 'dataset\santander-value-prediction-challenge\split_eval.csv'. | Industry | Predicting transaction values, regression, single table, 5k columns, suitable for complex algorithms | Regression | 3 | RMSLE |

