# OpenReview forum: "Data Interpreter: An LLM Agent For Data Science"
_ICLR.cc/2025/Conference — Submitted to ICLR 2025_

### Official Review · Reviewer_SPrw · 2024-10-26

**Soundness:** 3
**Presentation:** 2
**Contribution:** 2
**Rating:** 5
**Confidence:** 4

**Summary:**

The paper proposes Data Interpreter, an LLM agent agent designed to solve data science tasks in an end-to-end fashion. Compared to existing works on LLM agent, this work presents two innovative modules: (1) the Task Graph Generator, which employs Hierarchical Graph Modeling to break down tasks into smaller, more manageable components, and (2) the Action Graph Generator, which utilizes Programmable Node Generation to select or generate tools and code for execution. These enhancements result in a 19.01% improvement in accuracy over the naive GPT-4 model on the InfiAgent-DABench and demonstrate competitive performance across various benchmarks, surpassing existing approaches.

**Strengths:**

- The paper addresses an intriguing and timely problem, which is how human can utilize the ability of LLM to perform end-to-end tasks for users efficiently.
- The authors conduct an extensive series of experiments across various data science tasks, achieving strong benchmark performance, which highlights the potential and effectiveness of the proposed approach.

**Weaknesses:**

- Section 3 introduces the two primary contributions of the paper—Hierarchical Graph Modeling and Programmable Node Generation. However, some descriptions are way too vague and redundant, and should be streamlined and clarified. The current form leaves readers with many questions. For example:

  - Line 200-201: The nature of the relationship $r$ between processes lacks clarity. Is $r$ determined by predefined constraints that guide the LLM in constructing the **Directed Acyclic** Graph (DAG)? For example, does it ensure that steps like data visualization precede model training? Although introduced as a challenge, it’s later stated (line 214-215) that subprocesses exchange intermediate results and parameters, also represented by $r$. However, this raises questions: Wouldn’t exchanging information between processes that occur sequentially lead to cyclic dependencies in a DAG? How could a process that is supposed to appear later in a DAG exchange information with some other process that has already been executed? More importantly, under such a formulation, isn't the previously stated challenge easily solved, since every subprocess is defined to have a relationship with each other?

  - Equation 2 incurs more confusion. The presented equation only states the objective is to select the graph that maximizes the expected performance. It leaves unclear how the system generates potential graphs and evaluates their expected performance without prior execution. Is the system generating all possible graphs at once? Does the evaluation happen after the execution of the graph? Elaboration on these points would help clarify the approach.

  - The paper mentions "refining relationships between subprocesses" on line 216, but the actual refinement process is unclear. The only thing that's close to it is "Task Graph Refinement" on line 266, it appears to focus on editing task nodes rather than relationships. Further clarification on whether “subprocess” and “task” refer to the same entities, and on the refinement process, would enhance reader understanding.

  - The Algorithm 1 as presented, includes a variety of functions that are not well defined, which can make it difficult to interpret the intended functionality. From a reader’s perspective, it appears as a general nested-loop structure. More explicit descriptions of key functions would improve clarity.

  - How is the task graph generator on line 256 implemented?

  - What is the episodic memory defined on line 312? The episodic memory concept appears to leverage existing technology rather than representing an innovation within this work.

  - Implementation Details in the appendix also only contains a single figure (Fig.6) that dives into the detail. However, several elements in this figure, such as "think", "act" and "combinations" lack clear definitions, which may hinder comprehension. Expanding these descriptions would provide us with some understanding of the methodology.

  - The *lack of reproducibility* of this work (see below) further exacerbates this problem.

- Continuous monitoring and iterative update seems less innovative, as it resembles functionality already present in major LLM-serving platforms, like ChatGPT, which also support updates following unsatisfactory results. Lines 262-264 suggest that this work automates this process by notifying the LLM when code execution encounters issues, which, while functional, is trivial.


- On line 533, the author states that "Our framework continuously monitors data changes and adapts to dynamic environments through iterative task refinement and graph optimization". However, it is unclear how "data change" is defined or detected. For example, how does the system recognize modifications in the underlying data (such as a file update at a previously specified path)? Additional explanation on this would be beneficial.

- No code nor any reproducibility statement has been provided by the authors. Given that the LLM backends used in this work (e.g. gpt-4, gpt-4o) as well as the datasets utilized in this work are publicly available, offering a reproducible example or allow readers to conduct experiments on their own would be extremely valuable in showcasing the agent's capabilities. Due to the missing of the code, I am unable to independently verify the results.

As it stands, the paper comes across more as a vague, high-level technical report rather than a fully detailed academic submission, as it primarily outlines the design without delving into the implementation specifics or the rationale behind certain design choices. Consequently, the work fall short of the rigor expected for ICLR.

Other minor problems in writing:

- Line 228: missing spaces before data exploration

- Line 244: At a more granular level -> At finer granularity

- Appendix A: Diversity and complexity insufficient -> Insufficient diversity and complexity.

- Figure 6: Recomendation -> Recommendation

**Questions:**

See weaknesses for a plethora of major questions.

Other questions include:
- Line 419-427 and Table 1 suggest that the proposed approach may underperform when using GPT-4, which the authors attribute to limitations in handling long contexts, proposing that this could be addressed with LLM backends that support extended context lengths like GPT-4o. This raises a few questions: Would similar performance improvements be observed with other LLMs (e.g., GPT-4-Turbo with 128k context or Qwen-72B with 32k context), several of which have already been tested in the ablation studies? Additionally, it would be valuable to know the primary failure cases of GPT-4 on InfiAgent-DABench and understand how Data Interpreter effectively mitigates these challenges.

---

> ### Author Response · Authors · 2024-11-23
> **Thanks for your review! Authors' feedback [1/4].**
>
> We sincerely thank the reviewer for their thorough and constructive feedback. Below we address each point in detail:
>
> **1. W1：Line 200-201: The nature of the relationship rrr between processes is unclear. Is rrr guided by predefined constraints, such as ensuring data visualization precedes model training in the DAG?**
>
> We appreciate the question regarding task relationships. The nature of relationship r here refers to the data dependency. We do not rely on predefined constraints to determine relationships between processes. Instead, these relationships emerge naturally from the LLM's comprehensive understanding of data science workflows and best practices. The LLM naturally understands the typical sequence of data science tasks through its inherent knowledge. For instance, the LLM recognizes that exploratory visualization precedes model training, while performance visualization follows it.
>
> To clarify, we will revise Lines 225-226 from "without relying on pre-defined steps or tasks" to "without relying on pre-defined steps, tasks, or relationships."
>
> **2. W2: Wouldn't exchanging information between processes lead to cyclic dependencies in a DAG?**
>
> Thank you for raising this point. We should clarify that information flows in our system are strictly unidirectional. Each subprocess can access outputs (code and data) from previous subprocesses, but not vice versa, preserving the acyclic property of the DAG.
>
> **3. W3:  More importantly, under such a formulation, isn't the previously stated challenge easily solved, since every subprocess is defined to have a relationship with each other?**
>
> While decomposing a complex workflow into tasks reduces complexity, the challenge in data science tasks lies in their dynamic nature. For example, a feature engineering task may generate new data columns that affect downstream tasks like model training and evaluation. The key challenge is not just connecting tasks, but effectively capturing and understanding intermediate results as well as adapting to dynamically changing data dependencies. Data Interpreter addresses this through its task graph structure, which allows for explicit modeling and updates of these dependencies.
>
> **4. W4:Equation 2 is unclear. It leaves unclear how the system generates potential graphs and evaluates their expected performance without prior execution. Is the system generating all possible graphs at once? Does the evaluation happen after the execution of the graph?**
>
> Thanks for pointing out this concern. We clarify that our system does not generate all possible graphs upfront.
>
> 1. Our objective function aims to maximize the performance of data modeling or analysis tasks through graph-based task dependency representation. Data Interpreter does not generate all possible graphs upfront; instead, it employs a progressive and iterative approach that adapts to runtime outcomes.
>
> 2. Specifically, Data Interpreter initializes with a basic workflow (e.g., preprocessing → feature engineering → model training) and progressively refines the task graph based on execution feedback. For instance, when feature engineering generates new columns, the system updates the task graph to incorporate these new data features and their relationships into subsequent model training steps. Performance evaluation occurs after graph execution, ensuring adaptability to runtime outcomes.
>
> **5. W5: Refining Relationships Between Subprocesses. Further clarification on whether “subprocess” and “task” refer to the same entities, and on the refinement process, would enhance reader understanding.**
>
> Thanks for your question. Let us clarify these concepts and processes.
>
> 1. **"Subprocess" and "task"**: In our paper, these terms refer to the same entities, representing atomic operations in data science workflows.
>
> 2. **Refinement Process:** While Task Graph Refinement (Line 266) appears to focus on nodes, it actually handles both node updates and relationship refinement simultaneously. When the system refines a task's implementation (node update), it automatically triggers the refinement of relationships with connected tasks. This happens because our action graphs capture both the task's internal logic and its data dependencies with other tasks.
>
>     For example, When a model fails due to missing features in validation data, our system not only updates the model training task but also automatically establishes new dependency edges between validation data processing and feature generation tasks. These dependencies may be implicitly indicated or implemented in the code. This demonstrates how node modifications naturally lead to relationship refinement through our action graphs.

---

> ### Author Response · Authors · 2024-11-23
> **Thanks for your review! Authors' feedback [2/4].**
>
> **6. W6: More explicit descriptions of key functions would improve clarity.**
> >```initialize_graph()```: Prompts the LLM with user requirements to initialize the task graph, outlining the initial workflow structure.
>
> > ```select_task_node()```: Monitors task execution status and selects the next task node to execute . This includes marking completed tasks, retrieving related task execution results for the next task according to the task dependencies.
>
> > ```initialize_action_graph()```: Leverages the LLM to generate code snippets based on task descriptions, dependencies, and available tools.
>
> > ```execute()```: Executes the action graph (code) and returns runtime results, including execution outputs or traceback information in case of failure. Execution is performed in a Jupyter Notebook environment to support IPython code execution.
>
> > ```is_success()```: Extracts the runtime results from the notebook and detects whether there are errors or traceback messages. If no errors are found, the system marks the execution as successful; otherwise it's considered a failure.
>
> **7. W7: How is the task graph generator on line 256 implemented?**
>
> Please refer to the response to Reviewer 8RxQ W1, where we walk through a concrete example of how the system generates and refines task graphs during execution.
>
> **8. W8: What is the episodic memory defined on line 312?  The episodic memory concept appears to leverage existing technology rather than representing an innovation within this work.**
>
> Thanks for the question. In our work, episodic memory refers to the historical record of completed tasks, storing their code implementations, execution results and debugging process. When a task fails, our Data Interpreter utilizes this memory and current execution context to regenerate the code, leveraging graph structure to identify relevant historical tasks through their dependencies. While the concept of episodic memory itself is not new, our innovation lies in how we integrate it with the Data Interpreter for graph refinement.
>
> **9. W9: Implementation Details in the appendix also only contains a single figure (Fig.6) that dives into the detail. However, several elements in this figure, such as "think", "act" and "combinations" lack clear definitions.**
>
> Thank you for your comments. Here, "think," "act," and "combinations" refer to different stages of LLM calls responding to input text.
>
> 1. **"Think"**: refers to the initial reasoning phase where the LLM analyzes the input task description to: This step involves calling the LLM with the task description and user requirements to generate tool type descriptions and determine the related tools for solving the task.
>
> 2. **"Act"**: represents the tool selection and execution phase, In this step, the system processes the generation results to filter tools using methods such as BM25 or LLM-based ranking and executes the resulting code.
>
> 3. **"Combinations"**: As described in Lines 351-353, this leverages the LLM to synthesize a combination of selected tools, necessary helper functions, and connecting code into executable code snippets based on the task description and the defined tool schema.
>
> **10. W10/W13: The lack of reproducibility of this work (see below) further exacerbates this problem.**
>
> Please refer to the response for Reviewer 8RxQ Q3 for details on reproducibility.
>
>
> **11. W11: Innovation in Continuous Monitoring and Updates**
>
> Thank you for raising this point. We'd like to clarify that our approach goes beyond simple error handling and updates:
>
> 1. **Error-driven Refinement**: Beyond basic error handling, Data Interpreter uses execution feedback to guide task-specific debugging and regeneration. This is not just about notifying the LLM of errors, but about understanding and addressing the root causes in the data science context.
> 2. **Result-driven Optimization**: Even for successfully executed code, Data Interpreter analyzes numerical outputs to determine if additional validation or optimization tasks are needed. For example, when examining model performance metrics, the system may automatically introduce hyperparameter optimization tasks or additional validation steps to ensure result reliability.

---

> ### Author Response · Authors · 2024-11-23
> **Thanks for your review! Authors' feedback [3/4].**
>
> **12. W12: On line 533, the authors state that "Our framework continuously monitors data changes and adapts to dynamic environments through iterative task refinement and graph optimization." It is unclear, however, how "data change" is defined or detected, such as identifying file updates or modifications in the data.**
>
> Thank you for seeking clarification about data change detection. Data changes are detected in two scenarios:
>
> 1. **Explicit Data Transformations**: When tasks modify data (e.g., feature engineering creating new columns, preprocessing changing data distributions), Data Interpreter tracks these changes through action graph execution results and ensures transformations are consistent across tasks.
>
> 2. **Intermediate Results Analysis**: The system monitors task outputs for data characteristics that might require additional handling. For example, when analyzing sales trends, if seasonal patterns are detected, Data Interpreter automatically introduces analytical tasks to explore these temporal effects.
>
> Additionally, our framework utilizes code capabilities to locate and access files, enabling dynamic data loading during execution to detect source data changes.
>
> **13. W14: Other minor problems with writing**
>
> Thank you for catching these typos. We will address all writing issues in the revised version.
>
>
> **14. W15: As it stands, the paper comes across more as a vague, high-level technical report rather than a fully detailed academic submission, as it primarily outlines the design without delving into the implementation specifics or the rationale behind certain design choices. Consequently, the work fall short of the rigor expected for ICLR.**
>
> We appreciate the reviewer's feedback regarding the level of detail in the paper. While we understand these concerns, we would like to emphasize that our work makes non-trivial contributions, both conceptually and practically, that are well-supported by comprehensive validation.
> The core of our work lies in the **hierarchical graph modeling** approach, which directly addresses fundamental challenges in data science workflows through several key contributions:
>
> - **Graph-based divide-and-conquer for data science problems**: We propose representing the solution space as a Directed Acyclic Graph (DAG). The DAG decomposes complex problems into atomic, verifiable subproblems that are solved progressively in topological order. This divide-and-conquer strategy effectively reduces the search space and enhances the modularity and controllability of the problem-solving process.
> - **Step-wise optimization**: Instead of whole-program regeneration and re-verification whenever errors occur, our approach performs targeted local modifications. Specifically, by executing DAG nodes (tasks) incrementally, if a problem arises at a specific node, the optimization process is confined to the affected node and its subsequent dependencies, ensuring that preceding tasks or already completed solutions remain unaffected. This localized and step-wise approach significantly reduces computational overhead while improving solution efficiency.
> - **State-of-the-art performance**: We have demonstrated the framework's effectiveness through rigorous experiments across diverse and challenging data science scenarios, including data analysis, machine learning, mathematical reasoning and data-related open-ended tasks. Our methods achieve SOTA results in various tasks, and proved to be efficient in much more challenging scenarios such as Kaggle competitions and table answering.
>
>
> **15 Q1: The paper attributes GPT-4's underperformance to context length limits and suggests using longer-context models like GPT-4o. How do other long-context models perform in your experiments?**
>
> We appreciate the reviewer's questions about LLM context length and performance. Our recent experiments with DeepSeek-V2.5 (128k context, default temperature=0.0) directly address these concerns:
>
> The experimental results demonstrate that utilizing a larger context window enables agent frameworks to further enhance the performance of base LLMs on InfriAgent-DABench. This finding aligns with our previous observations using gpt-4-0613. Using DeepSeek-V2.5 as the foundation model, Data Interpreter shows significant improvements with a 4.11% increase in Question Accuracy compared to AutoGen and a 7.97% improvement over direct LLM invocation.
>
> | Framework/Model | Proportional Accuracy by Subquestions ⬆️| Accuracy by Questions ⬆️| Uniform Accuracy by Subquestions ⬆️|
> |----------------|---------------------------------------|---------------------|--------------------------------|
> | DeepSeek-V2.5 | 0.757 | 0.728 | 0.766 |
> | AutoGen (DeepSeek-V2.5) | 0.794 | 0.755 | 0.793 |
> | Data Interpreter (DeepSeek-V2.5) | 0.823 | 0.786 | 0.831 |

---

> ### Author Response · Authors · 2024-11-23
> **Thanks for your review! Authors' feedback [4/4].**
>
> **16. Q2: What are the specific failure cases of GPT-4 and how does Data Interpreter address them?**
>
> We demonstrate with a specific case (task_id:142) from InfriAgent-DABench that shows how Data Interpreter addresses GPT-4's limitations.
> ```json
> {
>     "id": 142,
>     "question": "Question 2: Is there a relationship between the difference in votes received by the Democratic and Republican parties and their percentage point difference?",
>     "concepts": [
>     "Correlation Analysis"
>     ],
>     "constraints": "Calculate the Pearson correlation coefficient (r) to assess the strength and direction of the linear relationship between the difference in votes and the percentage point difference. Assess the significance of the correlation using a two-tailed test with a significance level (alpha) of 0.05. Report the p-value associated with the correlation test. Consider the relationship to be linear if the p-value is less than 0.05 and the absolute value of r is greater than or equal to 0.5. Consider the relationship to be nonlinear if the p-value is less than 0.05 and the absolute value of r is less than 0.5. If the p-value is greater than or equal to 0.05, report that there is no significant correlation.",
>     "file_name": "election2016.csv",
>     "level": "hard"
> }
> ```
> **GPT-4 output:**
> ```python
> import pandas as pd
> from scipy.stats import pearsonr
> data = pd.read_csv('election2016.csv')
> data['vote_diff'] = data['votes_dem'] - data['votes_gop']
> data['percentage_point_diff'] = (data['percentage_of_vote_dem'] - data['percentage_of_vote_gop']) * 100  # Calculate the Pearson correlation coefficient
> r, p = pearsonr(data['vote_diff'], data['percentage_point_diff'])
> ```
> > ```KeyError: 'percentage_of_vote_dem'```
> ```
> # Check the columns of the data
> print(data.columns)
>
> # Calculate the difference in votes and the percentage point difference
> data['vote_diff'] = data['votes_dem'] - data['votes_gop']
> data['percentage_point_diff'] = (data['per_dem'] - data['per_gop']) * 100
>
> # Calculate the Pearson correlation coefficient
> r, p = pearsonr(data['vote_diff'], data['percentage_point_diff'])
> ```
>
> >```# Output```
> (0.3584054656763721, 7.64673829612848e-96)
>
>
> **Data Interpreter output and anlysis:**
> ```python
> import pandas as pd
> import numpy as np
>
> # Load the CSV file
> file_path = r'election2016.csv'
> data = pd.read_csv(file_path)
>
> # Display the first few rows of the dataframe to understand its structure
> print(data.head())
>
> # Summary statistics of the numerical columns
> print(data.describe())
>
> # Select numerical columns for further analysis
> numerical_data = data.select_dtypes(include=[np.number])
>
> # Display the correlation matrix
> print(numerical_data.corr())
> ```
> >```# Output``` Dataset description and statistical info
>
> ```python
> from scipy.stats import pearsonr
>
> # Convert 'diff' and 'per_point_diff' to numeric values
> data['diff'] = data['diff'].str.replace(',', '').astype(float)
> data['per_point_diff'] = data['per_point_diff'].str.replace('%', '').astype(float)
>
> # Calculate Pearson correlation coefficient and p-value
> r_value, p_value = pearsonr(data['diff'], data['per_point_diff'])
>
> # Determine the relationship type
> if p_value < 0.05:
>     if abs(r_value) >= 0.5:
>         relationship_type = "linear"
>     else:
>         relationship_type = "nonlinear"
> else:
>     relationship_type = "none"
>
> # Output the results
> print(f"@correlation_coefficient[{r_value:.2f}] @p_value[{p_value:.4f}] @relationship_type[{relationship_type}]")
> ```
> >```# Output:```
> @correlation_coefficient[0.02] @p_value[0.1704] @relationship_type[none]
>
>
> **Limitations of GPT-4**:
> - Column misidentification: GPT-4 incorrectly assumed the presence of `percentage_of_vote_dem` and `percentage_of_vote_gop`, leading to a KeyError.
> - Lack of data validation: It did not verify column names or data formats prior to computation, causing avoidable errors.
> - Missed broader statistical context
> - Oversimplified analysis: GPT-4 calculated a correlation coefficient (r=0.358) but ignored critical steps like validating relationship type or accounting for data context.
>
> **How Data Interpreter resolves these issues**:
> - Started with exploratory data analysis (EDA)
> - Comprehensive data validation
> - Systematic approach to data exploration and analysis
>
> **Conclusion**
>
> GPT-4’s limitations in validating data, handling preprocessing, and contextualizing results led to incorrect conclusions. In contrast, Data Interpreter enhances reasoning effectiveness and output code quality through comprehensive data exploration and systematic analysis, providing robust context for its tasks.

---

> > ### Comment · Reviewer_SPrw · 2024-11-25
> > **Thank you very much for your detailed reply!**
> >
> > I appreciate the authors' very detailed responses, especially the examples provided and efforts to enhance the reproducibility of their work. I encourage the authors to incorporate many of these clarifications into the main text and appendix to improve the paper's overall readability. While several points of confusion have been addressed, some concerns remain unresolved:
> >
> > - W1-W5 & W7:
> > With the example provided in response to Reviewer 8RxQ W1, I now better understand the task graph generation process. However, the description in Section 3.1 seems inaccurate due to the following issues:
> > a. The task graph generation heavily relies on prompt engineering, which is not clearly conveyed.
> > b. There is no explicit mechanism to detect violations of the DAG constraint.
> > c. The objective in Eq. 2 does not appear to be explicitly optimized. Revise to the graph seems to only happen when the agent receives error.
> > I think this section requires significant revision to accurately depict the task planning process.
> >
> > - W6. Thank you for the clarification. The explanation provided in the response is clearer and more concise than Algorithm 1 in the text. I recommend replacing Algorithm 1 with this improved explanation.
> >
> > - W8. The authors define "episodic memory" as a "historical record of completed tasks, ... and debugging process." However, this appears analogous to the chat history functionality of major online personal assistants, such as ChatGPT, without substantial modifications. Given the significant challenge of context-length limitations in LLM-based agents, I am curious whether mechanisms beyond prompt engineering have been implemented to mitigate these constraints. Correct me if I'm wrong, based on the code provided, it seems this issue is addressed solely through prompt engineering.
> >
> > - W15. I appreciate the authors' clarification regarding the main contribution of this paper. Nevertheless, applying divide-and-conquer or modeling the problems as a graph isn't novel in the field of LLM Agent. As the author mentioned on line 214, GPTSwarm (Zhuge et. al ICML 2024) also represents language agents as graphs and performs step-level optimization. Could the authors elaborate on how this work differs from GPTSwarm beyond the application domain?

---

> > > ### Author Response · Authors · 2024-11-26
> > > **Thank you for your reply! Authors' feedback[2/2]**
> > >
> > > **5. Episodic memory implementation**
> > >
> > > Thanks for the question.
> > >
> > > In our work, episodic memory refers to the historical record of completed tasks, but unlike traditional chat history, it is structured within a task graph. This task-centric approach retains only essential details (e.g., instructions, dependencies, code, results), ensuring efficiency and relevance while addressing context-length constraints. For failed tasks, multiple execution results and corresponding code are preserved temporarily, but once completed or refined, the debugging memory is cleared. During refinement, only dependencies, code, and execution results from the task graph are utilized, maintaining a compact and task-focused context.
> > >
> > > We conducted an ablation study to compare the use of episodic memory with traditional chat history. The results, shown in Table 1, highlight the performance difference, with episodic memory consistently outperforming chat history across tasks due to its structured, task-centric approach.
> > >
> > > Table 1. Performance Comparison (*NPS indicates Normalized Performance Score*)
> > >
> > > | **NPS⬆️** | **Titanic** | **ICR** | **SCTP** |
> > > |----------------------------------|-------------|----------|----------|
> > > | **w/ episodic memory**           | 0.82        | 0.91     | 0.89     |
> > > | **w/ chat history**              | 0.81        | 0.88     | 0.84     |
> > >
> > >
> > >
> > >
> > > **6.  Compare with GPTSwarm, could the authors elaborate on how this work differs from GPTSwarm beyond the application domain?**
> > >
> > > Thanks for the valuable suggestions. Compared to GPTSwarm (Zhuge et. al ICML 2024), we model the LLM agent solving process as a network/graph and provide a detailed comparison in the table below:
> > >
> > > | Method            | Graph structure           | Node                                      | Edge (Connection)                               | Optimization       | Feedback                           | Tool               | Optimizer            | Domain                  |
> > > |--------------------|---------------------------|-------------------------------------------|------------------------------------------------|--------------------|------------------------------------|---------------------|----------------------|----------------------
> > > | **GPTSwarm**      | Single-level flat graph   | Homogeneous prompt-based nodes           | Communication channels                         | Separately optimized | Scalar (objective metrics)         | --                  | RL                   |General-purpose problem solving|
> > > | **Data Interpreter** | Two-level hierarchical  | Task nodes: high-level planning; Action nodes: executable code | Task-level dependencies; Action-level data flow | Joint optimization | Text (code execution & validation metrics) | Specialized ML tools | Iterative refinement |Data science problems
> > >
> > >
> > > - **Joint optimization**: Data Interpreter jointly optimizes both nodes and edges, effectively addressing their interdependencies, while GPTSwarm supports only separate optimization of nodes or edges. This joint optimization approach is crucial in dynamic workflows where changes in edges (e.g., introducing new dependencies through a feature engineering step) often require updates to nodes (e.g., add corresponding processing operations to the datasets downstream). Our joint optimization strategy captures these interdependencies, enabling more robust overall graph optimization and execution.
> > >
> > > - **Feedback mechanism**: Data Interpreter uses execution feedback in the form of informative, actionable textual feedback, while GPTSwarm relies solely on scalar feedback. Scalar feedback provides only a quantitative measure of performance but lacks context for diagnosing errors. Textual feedback highlights specific issues within the graph and offers actionable guidance for refinement.
> > >
> > > - **Optimization Methodology**: Data Interpreter uses real-time iterative refinement, avoiding the costly offline training required by GPTSwarm's RL-based approach, making it more efficient and practical.

---

> > > ### Author Response · Authors · 2024-11-27
> > >
> > > Dear Reviewer SPrw,
> > >
> > > Thank you for taking the time to review our paper and for providing valuable feedback! We have carefully addressed your constructive comments and revised the manuscript to make the paper more comprehensive and robust. If you have any further questions or concerns, please do not hesitate to reach out so we can address them before the discussion period concludes.
> > >
> > > If you feel our responses have satisfactorily addressed your concerns, we would be truly grateful if you could consider raising your score to reflect that the issues have been resolved.
> > >
> > > Thank you again for your time and thoughtful review!
> > >
> > > Sincerely,
> > >
> > > The Authors

---

> > > > ### Comment · Reviewer_SPrw · 2024-11-27
> > > >
> > > > Thank the authors for the additional experiments conducted and the rewriting effort. I have raised my score to 5 to reflect my assessment to the revised version of this paper.

---

> > > > > ### Author Response · Authors · 2024-12-02
> > > > >
> > > > > Dear Reviewer SPrw,
> > > > >
> > > > > Thank you for taking the time to carefully review the updates and for providing your thoughtful response. We sincerely appreciate your detailed consideration and your decision to update the score.
> > > > >
> > > > > If you have any further concerns, please feel free to let us know.
> > > > >
> > > > >
> > > > > Thank you once again for your valuable feedback and for the effort you have put into reviewing our work.
> > > > >
> > > > > Best regards,
> > > > >
> > > > > Authors

---

> ### Author Response · Authors · 2024-11-26
> **Thank you for your reply! Authors' feedback[1/2]**
>
> We appreciate the reviewers' careful examination and feedback, we would like to clarify and discuss the details.
>
> **Response to W1-W5 & W7:**
>
>  **1. Q: Regarding prompt engineering and task planning**
>
> We would like to clarify the following points regarding the task graph generation process:
>
> - **Use of LLM reasoning capabilities**:
>
> As evidenced in Line 199-200: *"Leveraging the reasoning ability of LLMs for general task decomposition, our method decomposes the solving process..."* and Line 225-226: *"Data Interpreter utilizes LLMs to perform task planning, providing only the project requirement as the goal without relying on pre-defined steps or tasks"*, our approach primarily relies on LLM reasoning capabilities. To improve clarity, we have revised the paper and added detailed prompts in Appendix E1 and E2, showing how concise prompts guide task decomposition with minimal reliance on prompt engineering.
>
> - **Graph data structure**
>
> The assertion that our method "heavily relies on" prompt engineering is not entirely accurate. The prompts used in our method are neither overly prescriptive nor lengthy instructions, and notably, they do not include any few-shot learning examples. Instead, we maintain a graph data structure to organize the generated reasoning outputs, which enforces dependencies between tasks and enables efficient task planning and refinement.
>
> We believe these revisions provide a clearer depiction of the role of LLMs and prompt engineering in our approach. Thank you for helping us improve the paper.
>
> **2. Q: Regarding DAG constraints**
>
> We understand the concern and acknowledge the usefulness of constraint checking mechanisms. To assess the necessity, we analyzed the task graphs generated on the InfriAgent-DABench, ML-Benchmark, and DS-Bench. We found **0** cycles in the generated graphs in these experiments. This suggests that the LLM's strong domain knowledge in data science tasks inherently guides it to avoid creating cyclic dependencies during task planning.
>
> Therefore, explicit cycle detection may not be strictly necessary in our current implementation. However, we recognize that incorporating a cycle detection mechanism could enhance robustness, especially in scenarios involving less structured domains or more complex task dependencies. We have revised the manuscript to include this in the limitation section. Moving forward, we will consider integrating such mechanisms in future work to further improve the system's adaptability and reliability.
>
> **3. Q: Regarding optimization of Equation 2**
>
> Thank you for your valuable suggestion. Our design leverages execution feedback, derived from runtime errors and LLM evaluations, to iteratively refine the task and action graphs, enhancing execution efficiency and overall performance. We will revise Eq. 2 to explicitly reflect this iterative optimization process based on feedback.
>
> The modified objective is:
>
> $$
> G^* = \arg\max_G \mathbb{E}{x\sim D} [\text{Performance}(G({p_i(x)}^n{i=1}, \mathcal{R}), F(G))]
> $$
>
> where:
>
> - \(F(G)\) represents the feedback mechanism that refines the graph \(G\) based on execution results, such as runtime errors or evaluations by LLMs.
> - $\\text{Performance}\$ measures the effectiveness of the graph after applying the feedback-driven adjustments, which reflects the practical utility and execution quality of the refined graph, ensuring it achieves both workflow success rate and quality metrics in handling complex workflows.
>
> This revised formulation explicitly integrates the feedback loop into the optimization process, highlighting that the graph topology is refined iteratively through feedback to improve execution efficiency and overall performance.
>
> We would appreciate your thoughts on whether this revision adequately addresses your concerns about the optimization process.
>
> **4. W6 and the explanation**
>
> Thank you for this helpful suggestion. We have revised Algorithm 1 to better reflect the practical implementation and make it more concise.

---

### Official Review · Reviewer_LELG · 2024-10-28

**Soundness:** 3
**Presentation:** 3
**Contribution:** 3
**Rating:** 5
**Confidence:** 4

**Summary:**

The paper introduces Data Interpreter, a LLM-based agent designed to automate complex data science workflows. The authors address the limitations of existing methods, which often struggle with long-term planning, dynamic data adjustments, and evolving task dependencies.

**Strengths:**

1. The combination of hierarchical graph modeling and programmable node generation not only effectively addresses task decomposition and code generation but also enhances system flexibility and adaptability through dynamic node generation and graph optimization.
2. The abstraction of tasks and actions into graph structures is rational, which allows for more granular control and flexibility in the execution process, making it easier to handle complex and interconnected tasks.
3. The experimental results show that Data Interpreter achieves significant performance improvements in these tasks.
4. How to use LLM agents to solve complex data science tasks has broad practical applications.

**Weaknesses:**

While the paper presents a compelling and innovative approach to solving data science tasks using a LLM agent, there are several areas where the work could be improved. Below is a detailed assessment of the weaknesses, with a specific focus on the experimental setup and the lack of token usage comparison.
1. One significant weakness in the experimental setup is the lack of a detailed comparison of token usage. The paper does not provide information on how many tokens are required for different tasks and datasets, nor does it compare the token consumption of Data Interpreter with other methods or baselines.
2. The authors should conduct a thorough analysis of token consumption for different tasks and methods.  Provide a comparative analysis of token usage between Data Interpreter and existing methods or baselines. This will help readers understand the efficiency and resource requirements of Data Interpreter relative to other solutions.
3. By addressing the lack of token usage comparison, the authors can provide a more comprehensive and insightful evaluation of Data Interpreter, making the paper more robust and valuable to the research community.

**Questions:**

Could you provide a detailed breakdown of the token consumption for different types of tasks and datasets, and compare it with existing methods or baselines? Understanding the token usage will help evaluate the efficiency and resource requirements of Data Interpreter, especially in resource-constrained environments.

---

> ### Author Response · Authors · 2024-11-23
> **Thanks for your review! Authors' feedback [1/1].**
>
> We sincerely thank the reviewer for their comprehensive feedback and the opportunity to clarify the efficiency of Data Interpreter.
>
> **W1/W2/W3/Q1: Comparison on token usage and time cost.**
>
> We compared our token cost (average per task) and inference time (average per task) across the ML-Benchmark, Open-ended Task Benchmark, MATH Dataset, and InfriAgent-DABench, while also reporting our performance. Our framework demonstrates **a state-of-the-art performance with competitive efficiency**.
>
> Table 1: **Performance and cost on ML-Benchmark**
> | Framework | Avg. Cost ($) ⬇️ | Avg. Inference Time (s) ⬇️ | Avg. Comprehensive Score ⬆️ |
> |------------|:--------------:|:---------------------:|:------------------------:|
> | AutoGen | 0.32 | 174.07 | 0.86 |
> | OpenInterpreter | 0.21 | 190.73 | 0.77 |
> | OpenDevin | 3.01 | 187.80 | 0.88 |
> | TaskWeaver | 0.37 | 168.55 | 0.69 |
> | XAgent | 20.09 | 6186.00 | 0.45 |
> | Data Interpreter | 0.84 | 237.31 | 0.95 |
>
>
> Table 2: **Performance and cost on Open-ended task benchmark** (Evaluated on OCR/WSC/WPI/IBR tasks)
> | Framework | Avg. Cost ($) ⬇️ | Avg. Inference Time (s) ⬇️ | Avg. Comprehensive Score ⬆️ |
> |------------|:--------------:|:---------------------:|:------------------------:|
> | AutoGen | 0.30 | 90.05 | 0.645 |
> | OpenInterpreter | 0.15 | 103.00 | 0.540 |
> | OpenDevin | 1.41 | 156.50 | 0.658 |
> | Data Interpreter | 0.34 | 117.25 | 0.953 |
>
>
> Table 3: **Performance and cost on MATH Dataset**
> | Framework | Avg. Cost ($) ⬇️ | Avg. Inference Time (s) ⬇️ | Avg. Accuracy ⬆️ |
> |------------|:--------------:|:---------------------:|:---------------:|
> | AutoGen | 0.242 | 120.99 | 0.500 |
> | Data Interpreter | 0.336 | 211.57 | 0.633 |
>
> Table 4: **Performance and cost on InfriAgent-DABench**
> | Framework | Avg. Cost ($) ⬇️ | Avg. Inference Time (s) ⬇️ | Avg. Accuracy ⬆️ |
> |------------|:--------------:|:---------------------:|:---------------:|
> | AutoGen (GPT-4o) | 0.112 | 42.42 | 88.72 |
> | AutoGen (GPT-4-0613) | 0.212 | 45.69 | 71.49 |
> | Data Interpreter (GPT-4o) | 0.017 | 49.44 | 94.93 |
> | Data Interpreter (GPT-4-0613) | 0.311 | 51.09 | 73.55 |
>
> **Analysis**
>
> 1) On ML-Benchmark (See Table 1), Data Interpreter achieves the highest comprehensive score (0.95) among all frameworks, though with moderate cost (0.84 USD) and inference time (237.31s) as shown in Table 1. While frameworks like OpenInterpreter achieve lower costs (0.21 USD) through one-time code generation, they show inferior performance (0.77).
>
> 2) In Table2, for open-ended tasks (See Table 2), Data Interpreter significantly outperforms baselines with a comprehensive score of 0.953, maintaining reasonable cost (0.34 USD) compared to OpenDevin (1.41 USD) and AutoGen (0.30 USD).
>
> 3) On specific domains like MATH Dataset (See Table 3) and InfriAgent-DABench (See Table 4), Data Interpreter consistently shows superior accuracy (63.3% and 94.93% respectively) while maintaining competitive efficiency, as demonstrated in Table 3 & 4. Notably, on InfriAgent-DABench, our approach achieves better performance with lower cost (0.017 USD vs. 0.112 USD) compared to AutoGen.
>
> 4) This performance advantage stems from our focus on dependency management, which helps avoid the redundancy and overthinking issues seen in frameworks like XAgent (which shows high cost of 20.09 USD and long inference time of 6186s in ML-Benchmark). Additionally, while previous state-of-the-art agent (i.e., AutoGen) occasionally suffers from uncontrolled dialogue loops leading to ineffective conversations, Data Interpreter maintains stable performance across different tasks and benchmarks.

---

> ### Author Response · Authors · 2024-11-26
>
> Dear Reviewer LELG,
>
> Thank you for taking the time to review our paper and for providing valuable feedback! We have carefully addressed your constructive comments and included the cost analysis in the appendix of our revised manuscript to make the paper more comprehensive and robust. If you have any further questions or concerns, please do not hesitate to reach out so we can address them before the discussion period concludes.
>
> If you feel our responses have satisfactorily addressed your concerns, we would be truly grateful if you could consider raising your score to reflect that the issues have been resolved.
>
> Thank you again for your time and thoughtful review!
>
> Sincerely,
>
> The Authors

---

> ### Author Response · Authors · 2024-12-02
>
> Dear Reviewer LELG,
>
> Thank you for your time and effort in reviewing our paper, as well as for your constructive feedback and valuable questions. We sincerely appreciate the thoughtfulness you have brought to the review process.
>
> As the rebuttal period concludes today, we kindly ask whether our responses have met your expectations or if further clarifications are needed. If our responses address your concerns, we would greatly appreciate your reconsideration of the score. Otherwise, we are happy to provide any additional clarifications within the remaining time.
>
> Thank you again for your valuable input, which has significantly contributed to improving our work.
>
> Best regards,
>
> Authors

---

### Official Review · Reviewer_2zbj · 2024-11-01

**Soundness:** 2
**Presentation:** 3
**Contribution:** 2
**Rating:** 5
**Confidence:** 3

**Summary:**

This paper introduces a multi-agent framework for the automated resolution of data science problems. Specifically, a hierarchical graph modeling module is designed to decompose complex problems into interconnected sub-problems, followed by a programmable node generation technique that addresses each sub-task through iterative code generation and refinement. Their approach (named Data Interpreter) achieves better results when compared with other SOTA models on three different evaluation datasets.

**Strengths:**

1. This work establishs a novel data science scaffold that interprets tasks as graph, iteratively refining both sub-task construction and execution. They test the effectiveness of this method and achieved better performance given some baselines, additionally, ablation studies also confirm the effectiveness of the proposed module.
2. This paper is well-structured and presents a clear workflow of their proposed method, and show some interesting demonstrations to validate their advancedness of this scaffold.

**Weaknesses:**

From my perspective, the incremental contribution of this paper appears modest and limited, for the following reasons:
1. Like other LM-based agents[1], this method also relies on prompting techniques and without introducing interesting or advanced reasoning capabilities. Thus, the incremental contribution may not align with the standards expected at such a top conference.
2. Although the experiments display substantial performance improvements, the tasks appear carefully curated and too easy, potentially making them less convincing as evidence of the method’s advancedness.
3. The paper lacks comparisons with recent benchmark results within the same domain, as seen in [1][3], which could have strengthened the empirical evaluation.

**Questions:**

The baseline tasks and methods selected for comparison seem outdated and relatively easy. Could you provide additional evaluations against more recent benchmarks (e.g., the tasks in [2][4]) to more assess the effectiveness of this method?  Are there particular reasoning techniques or capabilities you think would significantly advance the field if incorporated?\
\
\
\
[1] Dominik Schmidt, Zhengyao Jiang, and Yuxiang Wu. "Introducing Weco AIDE", URL: https://www.weco.ai/blog/technical-report, 2024.\
[2] Qian Huang, Jian Vora, Percy Liang etc.. "MLAgentBench: Evaluating Language Agents on Machine Learning Experimentation", ICML24.\
[3] Liqiang Jing, Zhehui Huang etc.. "DSBench: How Far Are Data Science Agents to Becoming Data Science Experts?", 2024.\
[4] Jun Shern Chan and Neil Chowdhury etc. "MLE-bench: Evaluating Machine Learning Agents on Machine Learning Engineering", 2024.

---

> ### Author Response · Authors · 2024-11-22
> **Updated contributions and additional comparisons on MLE-Bench [1/2]**
>
> We sincerely appreciate the reviewer's thoughtful feedback regarding our work's contribution and evaluation. We address each point while providing additional context and clarification.
>
> **1. W1: The novelty and contributions of the paper**
>
> We would like to highlight that our contribution is three-fold:
>
> - **Graph-based divide-and-conquer for data science problems**: Unlike existing methods that generate entire code solutions (e.g., AutoGen, OpenDevin, TaskWeaver, AIDE), which leads to a vast solution space. We propose representing the solution space as a Directed Acyclic Graph (DAG). The DAG decomposes complex problems into atomic, verifiable subproblems that are solved progressively in topological order. This divide-and-conquer strategy effectively reduces the search space and enhances the modularity and controllability of the problem-solving process.
>
> - **Step-wise optimization**: Instead of whole-program regeneration and re-verification whenever errors occur, our approach performs targeted local modifications. Specifically, by executing DAG nodes (tasks) incrementally, if a problem arises at a specific node, the optimization process is confined to the affected node and its subsequent dependencies, ensuring that preceding tasks or already completed solutions remain unaffected.This localized and step-wise approach significantly reduces computational overhead while improving solution efficiency.
>
> - **State-of-the-art performance**: Our extensive experiments validate these innovations. Our method achieves strong solution quality with 0.95 comprehensive score on ML-Bench and 25% improvement on InfriAgent-DABench, compared to the existing baselines (as shown in Table 1 and Table2). On MLE-Bench, it achieves comparable performance to AIDE while reducing overall computation time from 24 hours to around 800s.
>
> **2. W2/W3/Q1: The paper lacks comparisons with recent benchmark (e.g., MLE-Bench and DS-Bench).**
>
> We have conducted extensive additional experiments on MLE-Bench and DS-Bench as suggested.
>
> **MLE-Bench**
>
> Following Jimenez et al. [1], we crafted MLE-Bench-Lite for effective evaluation, which consists of 8 randomly sampled tasks from MLE-Bench. We evaluated these tasks with tighter time constraints (max 3h (Ours) vs original 24h (AIDE and OpenDevin) per task):
>
> > Experimental Setup:
> > - Hardware: 1 GPU (24G), Memory: 125G, CPU: 36 Core
> > - Time constraint: 3 hours per task (vs original 24h)
> > - LLM: gpt-4o-2024-08-06, temperature=0
>
> Table 1: **Performance Comparison**:
> | Task | AIDE (~24h/task) | OpenDevin (<24h) | Data Interpreter (<900s/task) |
> |:-----|:------------:|:-----------:|:----------------------:|
> | spooky-author-identification | **0.4533** | 0.5894 | 0.7338 ± 0.1311 |
> | random-acts-of-pizza | 0.6227 | 0.5918 | **0.6312 ± 0.0118** |
> | nomad2018-predict-conductors | **0.0636** | 0.1835 | 0.0663 ± 0.0005 |
> | aerial-cactus-identification | **0.9998** | 0.8728 | 0.9993 ± 0.0007 |
> | leaf-classification | **0.6729** | 0.9021 | 0.6749 ± 0.0891 |
> | dog-breed-identification | 5.4768 | 2.8599 | **1.0596 ± 0.0712** |
> | dogs-vs-cats-redux | 0.8993 | 0.3867 | **0.1094 ± 0.0013** |
> | detecting-insults | NA | **0.8678** | 0.5110 ± 0.1156 |
>
> Table 2: **Efficiency of Data Interpreter**:
> | Task | Time (s) ⬇️ | Cost ($) ⬇️|
> |:-----|:--------:|:--------:|
> | spooky-author-identification | 200.85 | 0.09 |
> | random-acts-of-pizza | 294.97 | 0.25 |
> | nomad2018-predict-conductors | 477.23 | 0.16 |
> | aerial-cactus-identification | 266.75 | 0.05 |
> | leaf-classification | 347.32 | 0.17 |
> | dog-breed-identification | 407.07 | 0.27 |
> | dogs-vs-cats-redux | 820.54 | 0.12 |
> | detecting-insults | 802.92 | 1.81 |
> | Avg.|452.21|0.37|
>
> **Analysis of experimental results**
>
> As shown in Table 1, despite significantly reduced time budget (>800s (Ours) vs. >=24h (AIDE and OpenDevin)), Data Interpreter demonstrates strong performance with 87.5% win rate compared to OpenDevin and surpasses AIDE in 3 tasks. All tasks were successfully completed within 452.21s in average (See Table 2), compared to baselines that require up to 24 hours, validating the efficiency of our approach.
>
> [1] Jimenez C E, Yang J, Wettig A, et al. Swe-bench: Can language models resolve real-world github issues?[J]. arXiv preprint arXiv:2310.06770, 2023.

---

> > ### Comment · Reviewer_2zbj · 2024-11-25
> > **Thansk for you reply**
> >
> > I deeply appreciate the authors’ detailed responses and experimental results. However, I am a bit confused about the results in Table 1 in your reply:
> > 1.  What are the evaluation metrics for these experiments? Do they include both MSE and accuracy? I noticed that some of the bolded metrics indicate ‘the larger, the better,’ while others suggest ‘the smaller, the better.’
> > 2.  Why does only your method report variance? Is it because the other methods were evaluated only once while yours underwent multiple trials? If so, is this comparison fair?
> > 3.  How is the 87.5% win rate calculated? It seems that your method achieved the best results only in tasks 2, 6, and 7, as far as I can see.

---

> > > ### Author Response · Authors · 2024-11-27
> > >
> > > Dear Reviewer 2zbj,
> > >
> > > Thank you for taking the time to review our paper and for providing valuable feedback! We have carefully addressed your constructive comments to make the paper more comprehensive and robust. If you have any further questions or concerns, please do not hesitate to reach out so we can address them before the discussion period concludes.
> > >
> > > If you feel our responses have satisfactorily addressed your concerns, we would be truly grateful if you could consider raising your score to reflect that the issues have been resolved.
> > >
> > > Thank you again for your time and thoughtful review!
> > >
> > > Sincerely,
> > >
> > > The Authors

---

> > > ### Author Response · Authors · 2024-12-02
> > >
> > > Dear Reviewer 2zbj,
> > >
> > > Thank you for your time and effort in reviewing our paper, as well as for your constructive feedback and valuable questions. We sincerely appreciate the thoughtfulness you have brought to the review process.
> > >
> > > As the rebuttal period concludes today, we kindly ask whether our responses have met your expectations or if further clarifications are needed. If our responses address your concerns, we would greatly appreciate your reconsideration of the score. Otherwise, we are happy to provide any additional clarifications within the remaining time.
> > >
> > > Thank you again for your valuable input, which has significantly contributed to improving our work.
> > >
> > > Best regards,
> > >
> > > Authors

---

> > > > ### Comment · Reviewer_2zbj · 2024-12-03
> > > > **Response to Rebuttal**
> > > >
> > > > Thank you to the authors for their efforts in conducting the additional experiments. I have also carefully reviewed the opinions of the other reviewers. While the additional experiments address some of the questions I previously raised, I still have concerns regarding the novelty and incremental contributions. Therefore, I will maintain my original score.

---

> > > > > ### Author Response · Authors · 2024-12-04
> > > > >
> > > > > Thank you for taking the time to carefully review the updates and for providing your thoughtful response. We sincerely appreciate your detailed consideration and respect your decision to maintain the score.
> > > > >
> > > > > If you have any further concerns, please feel free to let us know.
> > > > >
> > > > > Thank you once again for your valuable feedback and for the effort you have put into reviewing our work.
> > > > >
> > > > > Best regards,
> > > > >
> > > > > Authors

---

> ### Author Response · Authors · 2024-11-23
> **Thanks for your review! Authors' feedback [2/2].**
>
> **2. W2/W3/Q1: The paper lacks comparisons with recent benchmark (e.g., MLE-Bench and DS-Bench). (continued)**
>
> **DS-Bench**
>
> Following the evaluation setup of DS-Bench, we conducted our experiments as follows: For the data analysis task, we randomly sampled 64 tasks, and for the data modeling task, we randomly sampled 10 tasks.
>
> Table 3: **Performance on data analysis tasks**
> | Method | Model | Task-level Accuracy (%) ⬆️ | Competition-level Accuracy (%) ⬆️ | Total Cost ($) ⬇️| Inference Time (s) ⬇️|
> |:----------|:-------|:---------------------|:---------------------------|:--------------|:-----------------|
> | AutoGen | GPT-4o | 43.8 | 38.7 | 4.88 | 27.19 |
> | Data Interpreter | GPT-4o | 47.2 (+7.8%) | 43.8 (+13.2%) | 3.56(-27.1%) | 45.04(+65.65%) |
> | AutoGen | GPT-4o-mini | 40.6 | 34.7 | 0.47 | 33.19 |
> | Data Interpreter | GPT-4o-mini | 43.8 (+7.9%) | 42.9 (+23.6%) | 0.10 (-78.7%) | 14.54 (-56.2%) |
>
> Table 4: **Performance on data modeling tasks**
> | Method | Model | Task Success Rate (%) ⬆️ | RPG ⬆️| Total Cost ($) ⬇️ | Inference Time (s) ⬇️ |
> |:----------|:-------|:---------------------|:----|:--------------|:-----------------|
> | AutoGen | GPT-4o | 60.0 | 0.293 | 0.67 | 94.79 |
> | Data Interpreter | GPT-4o | 100.0 (+66.7%) | 0.538 (+83.6%) | 1.60(+138.8%) | 347.09(+266.2%) |
> | AutoGen | GPT-4o-mini | 20.0 | 0.160 | 0.01 | 15.71 |
> | Data Interpreter | GPT-4o-mini | 100.0 (+400.0%) | 0.487 (+204.4%) | 0.39(+3800.0%) | 280.02(+1682.4%) |
>
> **Analysis of experimental results**
>
> **State-of-the art performance**:
>
> 1) For data analysis task, table 3 demonstrate that Data Interpreter achieves significant improvements in competition-level accuracy, with a 13.2% improvement using GPT-4o and a 23.6% improvement using GPT-4o-mini, while reducing costs by 27.1% and 78.7% respectively. Notably, while Data Interpreter shows longer execution times due to its comprehensive statistical analysis and verification, this approach proves particularly effective for complex table analysis tasks.
>
> 2) For data modeling tasks, Data Interpreter achieves 100% task success rate across both GPT-4o and GPT-4o-mini, with significant RPG improvements of 83.6% and 204.4% respectively. Through iterative task graph refinement, Data Interpreter can self-correct and regenerate failed tasks to complete the modeling workflow. While this results in higher cost and inference time compared to AutoGen's simple strategy, it ensures superior task completion quality.
>
> **Better stability in various tasks**: In both DS-Bench (Table 3 & 4) and InfriAgent-DABench (Table 1 in the main paper), the previous state-of-the-art agent framework (i.e., AutoGen) occasionally faces uncontrolled dialogue loops in data analysis tasks, leading to prolonged inefficiency and increased costs, while Data Interpreter maintains stable performance.

---

> ### Author Response · Authors · 2024-11-26
> **Thank you for your reply! Authors' feedback[1/1]**
>
> Thank you for your careful review and insightful questions regarding our experimental results presented in Table 1. We appreciate the opportunity to clarify these important points.
>
> **R1: Thank you for point this out, we extend the metadata for each task to provide better clarity**
>
>  | Task | AIDE (~24h/task) | OpenDevin (<24h) | Data Interpreter (<900s/task) | Modality | Metric | Is_lower_better |
> |:-----|:----------------:|:----------------:|:----------------------------:|:---------|:--------|:---------------:|
> | spooky-author-identification | **0.4533** | 0.5894 | 0.7338 ± 0.1311 | Text | Multi Class Log Loss | TRUE |
> | random-acts-of-pizza | 0.6227 | 0.5918 | **0.6312 ± 0.0118** | Text | AUC | FALSE |
> | nomad2018-predict-conductors | **0.0636** | 0.1835 | 0.0663 ± 0.0005 | Tabular | RMSLE | TRUE |
> | aerial-cactus-identification | **0.9998** | 0.8728 | 0.9993 ± 0.0007 | Image | AUC | FALSE |
> | leaf-classification | **0.6729** | 0.9021 | 0.6749 ± 0.0891 | Image | Multi Class Log Loss | TRUE |
> | dog-breed-identification | 5.4768 | 2.8599 | **1.0596 ± 0.0712** | Image | Multi Class Log Loss | TRUE |
> | dogs-vs-cats-redux | 0.8993 | 0.3867 | **0.1094 ± 0.0013** | Image | Log loss | TRUE |
> | detecting-insults | NA | **0.8678** | 0.5110 ± 0.1156 | Text | AUC | FALSE|
>
>
> **R2: Thank you for this important question about variance reporting.**
>
> For AIDE and OpenDevin results, we directly report the results from MLE github repository[1]. All reported metrics represent averages across multiple experimental trials. We will update Table 1 with the standard deviations for the baseline methods (not included yet due to time limitation).
>
>
>
> **R3: Regarding the 87.5% win rate calculation**
>
> Sorry for the mistake. We miscalculated the result of 'spooky-author-identification' task. Comparing Data Interpreter (DI) with OpenDevin across all 8 tasks and accounting for whether metrics should be minimized or maximized, DI achieves better performance in 6 tasks (random-acts-of-pizza, nomad2018-predict-conductors, aerial-cactus-identification, leaf-classification, dog-breed-identification, and dogs-vs-cats-redux), yielding a win rate of 6/8 = 75%.
>
>
> [1]. https://github.com/openai/mle-bench/tree/main/runs

---

### Official Review · Reviewer_8RxQ · 2024-11-03

**Soundness:** 3
**Presentation:** 3
**Contribution:** 3
**Rating:** 6
**Confidence:** 4

**Summary:**

This paper introduces Data Interpreter, a solution for constructing and executing data analytical pipelines on a given dataset (to be analyzed) and task specification. The solution is based on a hierarchical graph modeling of complex task decompositions, from task graph generator, action graph generator, to graph executor. The abilities of LLMs, including reasoning, planning, and code generation, are utilized in different phases and composed as an end-to-end solution. Data Interpreter is evaluated on various benchmarks and datasets.

**Strengths:**

S1. A system with strong motivations and real use cases is built and presented.

S2. The paradigm of task deposition is reasonable and likely to utilize LLM's abilities in an effective way.

S3. The solution works for various types of tasks and shows good performance in experiments.

**Weaknesses:**

W1. Overall, the presentation of the overview and some details of the solution is clear. Better if there is a real and complete example in the paper to help explain intuitions and more details of the solution.

W2. Overhead of the solution (e.g., resource, time, and cost for each task) is not analyzed and reported.

**Questions:**

Q1. Better if there is an end-to-end example in Section 3, which help explain how different components work in Data Interpreter in more details and intuitions. For example, how a task graph is generated step by step, what happens if there is an execution failure, and so on.

Q2. Analyze and/or report the overhead of the solution (e.g., resource, time, and cost for each task) as well as different components.

Q3. Reproducibility of the experimental results need to be commented (how to ensure the reproducibility?)

---

> ### Author Response · Authors · 2024-11-22
> **Thanks for your review! Authors' feedback [1/2].**
>
> Thank you for your valuable feedback highlighting our work's strengths in benchmarking and experimental evaluation. We will address your questions in the following response.
>
> **1. W1 & Q1: Request for complete example showing step-by-step task graph generation and error handling**
>
> Thank you for this insightful comment. We agree on the importance of detailed examples to explain our approach.
>
> We use the *water pump failure prediction* (as shown in Figure 2) as a complete example to demonstrate: task graph generation process, action graph generation steps and error handling and refinement mechanisms.
>
> - Initial Task Graph Generation
>
> The Data Interpreter generates an initial task graph using the project requirement and prompt for planning as follows:
> ```markdown
> PLAN_PROMPT = """
> # Context:
> {context}
> # Available Task Types:
> {task_type_desc}
> # Task:
> Based on the context, write a plan or modify an existing plan of what you should do to achieve the goal. A plan consists of one to {max_tasks} tasks.
> If you are modifying an existing plan, carefully follow the instruction, don't make unnecessary changes. Give the whole plan unless instructed to modify only one task of the plan.
> If you encounter errors on the current task, revise and output the current single task only.
> Output a list of jsons following the format:
> [
>     {{
>         "task_id": str = "unique identifier for a task in plan, can be an ordinal",
>         "dependent_task_ids": list[str] = "ids of tasks prerequisite to this task",
>         "instruction": "what you should do in this task, one short phrase or sentence",
>         "task_type": "type of this task, should be one of Available Task Types",
>     }},
>     ...
> ]
> """
> ```
>
> Then, an initialized task graph with 7 tasks:
>
> ```
> [
>     {
>         "task_id": "1",
>         "dependent_task_ids": [],
>         "instruction": "Perform data loading and preliminary exploration of the train and eval datasets. Fill missing values and apply MinMax scaling.",
>         "task_type": "eda"
>     },
>     {
>         "task_id": "2",
>         "dependent_task_ids": [
>             "1"
>         ],
>         "instruction": "Conduct correlation analysis and provide descriptive statistics.",
>         "task_type": "eda"
>     },
>     {
>         "task_id": "3",
>         "dependent_task_ids": [
>             "1"
>         ],
>         "instruction": "Perform outlier detection using Isolation Forest to identify and handle anomalies.",
>         "task_type": "eda"
>     },
>     {
>         "task_id": "4",
>         "dependent_task_ids": [
>             "2"，
>             "3"
>         ],
>         "instruction": "Execute feature engineering, including General Selection, Target Mean Encoding, and Variance Based Selection to prepare features for model training.",
>         "task_type": "feature_engineering"
>     },
>     {
>         "task_id": "5",
>         "dependent_task_ids": [
>             "4"
>         ],
>         "instruction": "Split the data and train predictive models using Random Forest and XGBoost.",
>         "task_type": "model_train"
>     },
>     {
>         "task_id": "6",
>         "dependent_task_ids": [
>             "5"
>         ],
>         "instruction": "Evaluate the model's performance and generate an evaluation report.",
>         "task_type": "model_evaluate"
>     },
>     {
>         "task_id": "7",
>         "dependent_task_ids": [
>             "5",
>             "6"
>         ],
>         "instruction": "Visualize the analysis and prediction results, including classification reports and confusion matrix, and serialize the model.",
>         "task_type": "visualization"
>     }
> ]
> ```
> - Action Graph Generation: Data Interpreter utilizes LLMs to generate an action graph for each task.  *PLAN_STATUS* prompt maintains execution context and task graph state, while *ACTION_GRAPH* prompt generates executable code for each task node. The prompts are as follows:
>
> ````
> PLAN_STATUS = """
> ## Finished Tasks
> ### code
> ```python
> {code_written}
> ```
>
> ### execution result
> {task_results}
>
> ## Current Task
> {current_task}
>
> ## Task Guidance
> Write complete code for 'Current Task'. And avoid duplicating code from 'Finished Tasks', such as repeated import of packages, reading data, etc.
> Specifically, {guidance}
> """
>
> Action_Graph_Prompt = """
> # User Requirement
> {project_requirement}
>
> # Plan Status
> {plan_status}
>
> # Tool Info
> {tool_info}
>
> # Constraints
> - Take on Current Task if it is in Plan Status, otherwise, tackle User Requirement directly.
> - Ensure the output new code is executable in the same Jupyter notebook as the previous executed code.
> - Always prioritize using pre-defined tools for the same functionality.
>
> # Output
> While some concise thoughts are helpful, code is absolutely required. Always output one and only one code block in your response. Output code in the following format:
> ```python
> your code
> ```
> """
> ````

---

> ### Author Response · Authors · 2024-11-22
> **Thanks for your review! Authors' feedback [2/2].**
>
> A example of action graph for Task 4 is:
> ```python
> ...
> general_selection = GeneralSelection(label_col='machine_status')
>
> data_train = general_selection.fit_transform(data_train)
> data_eval = general_selection.transform(data_eval)
>
> # Apply TargetMeanEncoder to categorical columns
> encoder = TargetMeanEncoder(col=object_columns_train[0], label='machine_status')
>
> data_train = encoder.fit(data_train).transform(data_train)
> data_eval = encoder.transform(data_eval)
> ...
> ```
> - Error Handling and Refinement:
> When encountering task execution failures (e.g., Task 4 feature engineering), Data Interpreter employs step-by-step debugging with the *REFLECTION_PROMPT*, and attempts to regenerate codes.
> ````
> REFLECTION_PROMPT = """
> [example]
> Here is an example of debugging with reflection.
> {debug_example}
> [/example]
>
> [context]
> {context}
>
> [previous impl]:
> {previous_impl}
>
> [instruction]
> Analyze your previous code and error in [context] step by step, provide me with improved method and code. Remember to follow [context] requirement. Don't forget to write code for steps behind the error step.
> Output a json following the format:
> ```json
> {{
>     "reflection": str = "Reflection on previous implementation",
>     "improved_impl": str = "Refined code after reflection.",
> }}
> ```
> """
>
> ````
> After three failed action graph executions, Data Interpreter dynamically restructures the task graph: Tasks 1-3 remain unchanged, Task 4 is simplified to basic feature creation, a new Task 5 for feature selection is introduced, and subsequent tasks (e.g., original Task 5 becoming Task 6) are automatically reindexed with updated dependencies.
> ````
> ...
> {
>     "task_id": "4",
>     "dependent_task_ids": [
>             "2"，
>             "3"
>         ],
>     "instruction": "Create engineered features from sensor readings"，
>     "task_type": "feature_engineering"
> },
> {
>     "task_id": "5",
>     "dependent_task_ids": [
>             "4",
>         ],
>     "instruction": "Perform feature selection using statistical methods and importance analysis"，
>     "task_type": "feature_engineering"
> },
>  {
>         "task_id": "6",
>         "dependent_task_ids": [
>             "4",
>             "5"
>         ],
>         "instruction": "Train a predictive model to determine machine status"，
>         "task_type": "model_train"
>     },
>   ...
> ````
> **2. Q2: Overhead of the solution (e.g., resource, time, and cost for each task) is not analyzed and reported.**
>
> Thank you for raising the question about system overhead. Based on our evaluation of various tasks, here are the detailed overheads:
>
> - **ML-Benchmark**
> | Framework | Avg. Cost ($) ⬇️ | Avg. Inference Time (s) ⬇️ | Avg. Comprehensive Score ⬆️ |
> |------------|:--------------:|:---------------------:|:------------------------:|
> | AutoGen | 0.32 | 174.07 | 0.86 |
> | OpenInterpreter | 0.21 | 190.73 | 0.77 |
> | OpenDevin | 3.01 | 187.80 | 0.88 |
> | TaskWeaver | 0.37 | 168.55 | 0.69 |
> | XAgent | 20.09 | 6186.00 | 0.45 |
> | Data Interpreter | 0.84 | 237.31 | 0.95 |
>
>
> - **Open-ended task benchmark**
> | Framework | Avg. Cost ($) ⬇️ | Avg. Inference Time (s) ⬇️ | Avg. Comprehensive Score ⬆️ |
> |------------|:--------------:|:---------------------:|:------------------------:|
> | AutoGen | 0.30 | 90.05 | 0.645 |
> | OpenInterpreter | 0.15 | 103.00 | 0.540 |
> | OpenDevin | 1.41 | 156.50 | 0.658 |
> | Data Interpreter | 0.34 | 117.25 | 0.953 |
>
>
> - **MATH Dataset**
> | Framework | Avg. Cost ($) ⬇️ | Avg. Inference Time (s) ⬇️ | Avg. Accuracy ⬆️ |
> |------------|:--------------:|:---------------------:|:---------------:|
> | AutoGen | 0.242 | 120.99 | 0.500 |
> | Data Interpreter | 0.336 | 211.57 | 0.633 |
>
> - **InfriAgent-DABench**：*Cost and inference time are calculated per task*
> | Framework | Avg. Cost ($) ⬇️ | Avg. Inference Time (s) ⬇️ | Avg. Accuracy ⬆️ |
> |------------|:--------------:|:---------------------:|:---------------:|
> | AutoGen (GPT-4o) | 0.112 | 42.42 | 88.72 |
> | AutoGen (GPT-4-0613) | 0.212 | 45.69 | 71.49 |
> | Data Interpreter (GPT-4o) | 0.017 | 49.44 | 94.93 |
> | Data Interpreter (GPT-4-0613) | 0.311 | 51.09 | 73.55 |
>
> Notably, when using GPT-4o, AutoGen occasionally sufferes from uncontrolled dialogue loops, leading to prolonged ineffective conversations and increased costs, while Data Interpreter maintaines stable performance.
>
> **3. Q3. Reproducibility of the experimental results need to be commented (how to ensure the reproducibility?)**
>
> Thank you for raising this important point about reproducibility. We ensure reproducibility through:
> 1. Data & Evaluation
>   - All the datasets are publicly available with preprocessing scripts provided.
>   - Evaluation metrics & implementations are detailed in Appendix D.2.
> 2. Code Availability
>   - We will upload the source code to Supplementary Material, and a public GitHub repository will be released upon acceptance.
> 3. Experiment Setup
>   - Environment: Linux OS, 24GB GPU
>   - LLM Settings:
>     - ML-Benchmark & Open-ended Task: gpt-4-1106-preview (temperature=0)
>     - InfriAgent-DABench: gpt-4-0613 and gpt-4o (temperature=0)

---

### Comment · Area_Chair_AUbg · 2024-11-21
**No author response yet**

Dear Submission5663 Authors,

ICLR encourages authors and reviewers to engage in asynchronous discussion up to the 26th Nov deadline. It would be good if you can post your responses to the reviews soon.

---

> ### Author Response · Authors · 2024-11-24
> **Thank you for reminder!**
>
> Thank you for your reminder! We sincerely appreciate the valuable feedback from the reviewers, and we have updated the supplementary materials with our code and data. We have carefully considered and responded to each review comment in detail.

---

### Meta-Review · Area_Chair_AUbg · 2024-12-09

**Metareview:**

The paper proposes a system for automating data science problems using LLM agents.

Reviewers agreed that the problem being solved is significant, and that the task graph formulation at least is novel. Results on task accuracy and dollar cost-effectiveness are another strength of the paper.

However, multiple reviewers raised the same concern about novelty, in that no new reasoning techniques were proposed; ultimately the system relies on prompt engineering. While papers relying solely on prompt engineering might be a good fit for other conferences, state of the art approaches to LLM agents at conferences like ICLR would involve, for example, modifying the model or its decoding process, or proposing a new meta-optimization problem over the agents. The expectation is that the LLMs themselves should be modified, rather than being treated as black boxes.

**Additional Comments On Reviewer Discussion:**

The authors addressed reviewer concerns about task difficulty and missing benchmarks. However, the concern about a lack of innovation in LLM agent reasoning was not addressed in the rebuttal.

---

### Decision · Program_Chairs · 2025-01-22

Reject